# Immunometabolic determinants of long-term response in leukemia patients receiving CD19 CAR T cell therapy

Lior Goldberg [1,2] ✉, Eric R. Haas [1], Jiaqi Wu[1], Bryan Garcia[1], Ryan Urak[1], Vibhuti Vyas[1], Ruby Espinosa[1], Tamara Munoz[1], Shirley Bierkatz[1], Khyatiben V. Pathak[3,4], Nathaniel P. Hansen[3,4], Patrick Pirrotte [3,4], Jyotsana Singhal[5], James L. Figarola[5], Ricardo Zerda Noriega[6], Zhuo Li[6], Dasol Wi[7,8,9], Erin Tanaka[7,8], Ramon Klein Geltink [7,8,10,11], Min-Hsuan Chen[12], Xiwei Wu[12], Jamie R. Wagner[1], Jinny Paul[1], Mary C. Clark [1], Dat Ngo[1], Ibrahim Aldoss [1], Stephen J. Forman [1,13] & Xiuli Wang [1,13] ✉

Although most patients with relapsed/refractory B-cell acute lymphoblastic leukemia (B-ALL) receiving CD19-targeted chimeric antigen receptor (CAR) T cell therapy achieve remission, loss of CAR T cell functionality and subsequent relapse remains an unmet therapeutic need. Herein, we apply an integrative approach to study the immunometabolism of pre- and post-infusion CD19-CAR T cells of patients with relapsed/refractory B-ALL. Pre-infusion CAR T cells of long-term responders (LTR) have increased oxidative phosphorylation, fatty acid oxidation, and pentose phosphate pathway activities, higher mitochondrial mass, tighter cristae, and lower mTOR expression compared to products of short-term responders. Post-infusion CAR T cells in bone marrow (BM) of LTR have high immunometabolic plasticity and mTOR-pS6 expression supported by the BM microenvironment. Transient inhibition of mTOR during manufacture induces metabolic reprogramming and enhances anti-tumor activity of CAR T cells. Our findings provide insight into immunometabolic determinants of long-term response and suggest a therapeutic strategy to improve long-term remission.

CD19-targeted chimeric antigen receptor (CD19-CAR) T cell therapy has revolutionized treatment of relapsed/refractory (r/r) B-cell acute lymphoblastic leukemia (B-ALL). While most patients initially achieve remission, approximately 30-80% of patients experience CD19-positive relapse[1], highlighting the need to improve therapeutic efficacy. T cell metabolic fitness is critical for antitumor immunity, directly impacting T cell function, differentiation, and persistence[2], yet the metabolic features of CAR T cells have not been fully explored. Gaining

[1]Department of Hematology and Hematopoietic Cell Transplantation, T Cell Therapeutics Research Laboratories, Beckman Research Institute, City of Hope, Duarte, CA, USA. [2]Department of Pediatrics, City of Hope, Duarte, CA, USA. [3]Integrated Mass Spectrometry Shared Resource, City of Hope, Duarte, CA, USA. [4]Early Detection and Prevention Division, Translational Genomics Research Institute, Phoenix, AZ, USA. [5]Division of Diabetes and Metabolic Diseases Research, Beckman Research Institute, City of Hope, Duarte, CA, USA. [6]Core of Electron Microscopy, City of Hope, Duarte, CA, USA. [7]Department of Pathology and Laboratory Medicine, University of British Columbia, Vancouver, BC, Canada. [8]BC Children's Hospital Research Institute, Vancouver, BC, Canada. [9]Interdisciplinary Oncology Program, University of British Columbia, Vancouver, BC, Canada. [10]Department of Molecular Oncology, BC Cancer Research Centre, Vancouver, BC, Canada. [11]Edwin S.H. Leong Centre for Healthy Aging, University of British Columbia, Vancouver, BC, Canada. [12]Integrative Genomics Core, City of Hope and Beckman Research Institute, Duarte, CA, USA. [13]These authors contributed equally: Stephen J. Forman, Xiuli Wang. ✉e-mail: lgoldberg@coh.org; xiuwang@coh.org

a deeper understanding of the immunometabolism of CAR T cells may lead to the identification of strategies to enhance the effectiveness of adoptive cell therapy. Herein, we applied an integrative approach to study the immunometabolism of CD19-CAR T cells of patients with r/r B-ALL. Our multi-layer approach included functional metabolic assays, mitochondrial morphological evaluation, as well as bulk and single cell proteomic assays to elucidate the immunometabolic features of pre- and post-infusion CD19-CAR T cells. Insights from our findings suggest a potential metabolic therapeutic strategy to enhance the anti-tumor activity of CAR T cells in patients.

## Results

Our group previously conducted a phase 1/2 clinical trial evaluating the safety and efficacy of investigational CD19-CAR T cell therapy for adults with r/r B-ALL (NCT02146924)[3]. To study the immunometabolic characteristics associated with response to CAR T cell therapy, we investigated the pre-infusion CAR T cell products and biological samples from a cohort of sixteen patients treated in our clinical trial, all of whom responded to CD19-CAR T cell therapy. The cohort included two sub-groups ($n = 8$ per sub-group) based on length of response to CAR T cells, with short-term responders (STR) having CD19+ leukemia relapse within $142 \pm 30$ days following initial response and long-term responders (LTR) maintaining remission without subsequent therapy post-CAR T cell infusion. We pairwise matched the sub-groups, and both cohorts had similar demographic and clinical characteristics other than length of response, allowing us to study CAR T cell functionality without potential confounding variables such as antigen escape and post CAR T cell consolidation therapies (Table 1, Supplementary Data 1).

**Table 1 | Demographic and clinical characteristics of patients**

| Characteristic | Short-term responders (N = 8) | Long-term responders (N = 8) |
|---|---|---|
| Age at CAR T cell infusion | $46 \pm 6$ | $38 \pm 5$ |
| Sex – no. (%) | | |
| Female | 2 (25%) | 3 (37.5%) |
| Male | 6 (75%) | 5 (62.5%) |
| Mean pre-lymphodepletion blast % in bone marrow[a] | $22 \pm 12$ | $18 \pm 12$ |
| Number of CAR T cells infused | $200 \times 10^6$ | $200 \times 10^6$ |
| Days of CAR T cell manufacturing[a] | $13 \pm 0.3$ | $13 \pm 0.3$ |
| CAR T cell transduction efficiency[a] | $66 \pm 6$ | $78 \pm 5$ |
| Cytokine release syndrome max grade (Lee's criteria)—no. (%) | | |
| Grade 1 | - | 3 (37.5%) |
| Grade 2 | 6 (75%) | 4 (50%) |
| Grade 3 | - | - |
| Grade 4 | - | - |
| Neurotoxicity max grade (CTCAE v4.03)—no. (%) | | |
| Grade 1 | 4 (50%) | 4 (50%) |
| Grade 2 | 1 (12.5%) | 1 (12.5%) |
| Grade 3 | - | - |
| Grade 4 | - | - |
| Grade 5 | - | - |
| Complete remission at day 28 post infusion - no. (%) | 8 (100%) | 8 (100%) |
| Days between CAR T infusion and relapse[a] | $142 \pm 30$ | - |
| CD19 positive at relapse | 8 (100%) | 0 (0) |

[a]values are mean ± SEM.

### Pre-infusion products from LTR and STR have distinct immunometabolic profiles

We used an integrative approach to interrogate the immunometabolic properties of pre-infusion CAR T cell products from both sub-groups (Fig. 1a). Using the Seahorse metabolic functional assay, we determined that enriched CAR T cells from LTR have significantly higher basal and maximal oxidative phosphorylation (OXPHOS) without substantial change in basal glycolysis compared to those from STR (Fig. 1b-g). We then used mass cytometry (CyTOF), which allows for measurement of key metabolic transporters and enzymes as well as phenotypic and functional proteins[4], to study the immunometabolic profile at the single-cell proteomic level (Fig. 1h, Supplementary Fig. 1, Supplementary Data 2). We visualized single-cell data via principal component analysis (PCA) considering only the expression of proteins associated with cellular metabolism (GLUT1, HK, GAPDH, LDHA, G6PD, CD98, CS, ATP5A, CD36, CPT1A, ACADM, ACLY, FASN, SCD1, LAL, mTOR, pAKT, pS6, AMPK), and determined that CAR T cells from STR and LTR have distinct metabolic states (Fig. 1i). Overall, CD4 and CD8 CAR T cells from each sub-group showed similar respective biological trends in the expression of metabolic proteins (Fig. 1j). CD3 CAR T cells of LTR had significantly increased levels of HK, G6PD, and CPT1A compared to STR (Fig. 1k), suggesting that CAR T cells from LTR preferentially shunt HK glycolytic metabolites to the pentose phosphate pathway (PPP) rather than glycolysis, and by CPT1A could convert long-chain fatty acids into acylcarnitines for shuttling into the mitochondrial matrix for subsequent oxidation, previously associated with increased basal and maximal mitochondrial respiration[5–8]. Moreover, we found that for both CD4 and CD8 CAR T cells, HK and G6PD expression levels showed a significant correlation, further supporting the potentially coupled activity of the two step-limiting rate enzymes of glycolysis and the PPP, respectively (Fig. 1l). CD8 CAR T cells of LTR showed significantly decreased expression of the amino acid transporter CD98, which is associated with T cell effector activation[9], the lipolytic enzyme LAL, and the master metabolic regulator mTOR compared to those of STR (Fig. 1m). Lastly, we leveraged our single-cell mass cytometry approach (Supplementary Fig. 1) and compared the CAR-negative T cells in the pre-infusion products of both groups (Supplementary Fig. 2a). Similar to CD3 CAR-positive T cells, CD3 CAR-negative T cells of LTR exhibited significantly increased levels of HK, G6PD, and CPT1A compared to STR (Supplementary Fig. 2b), suggesting these immunometabolic changes are independent of CAR expression.

To complement our functional metabolism (Seahorse) and proteomic single cell analyses, we performed intracellular metabolomics analysis via untargeted mass spectrometry, which detected 703 non-redundant metabolites. Metabolic pathway enrichment analysis revealed that CAR T cell products of LTR have significantly increased pools of TCA metabolites, nucleotide biosynthesis intermediates, sphingolipid metabolism, and intermediates in central energy metabolism, as well as significantly decreased nucleotide catabolic and tRNA metabolism intermediates, which is critical for T cell activation[10], compared to those of STR (Fig. 1n, Supplementary Data 3a-c). These metabolomic data are consistent with our Seahorse and CyTOF analyses showing high OXPHOS, and increased expression of fatty acid oxidation and PPP enzymes in CAR T cells of LTR. Consistent with increased CPT1A expression (Fig. 1k) suggesting elevated fatty acid oxidation capacity, levels of both linoleic and oleic long-chain fatty acids were significantly lower while levels of acetyl and palmitoyl carnitines were significantly higher in CAR T cells of LTR compared to those of STR. Moreover, levels of citric and iso-citric acids were increased in CAR T cells of LTR vs. STR, which could facilitate increased TCA cycle activity and biosynthetic capacity, and levels of the nucleotide derivatives cytidine 5'-diphosphocholine, cytidine 5'-monophosphate, deoxyguanosine diphosphate as well as l-glutathione were increased in CAR T cells of LTR, which is supportive of increased

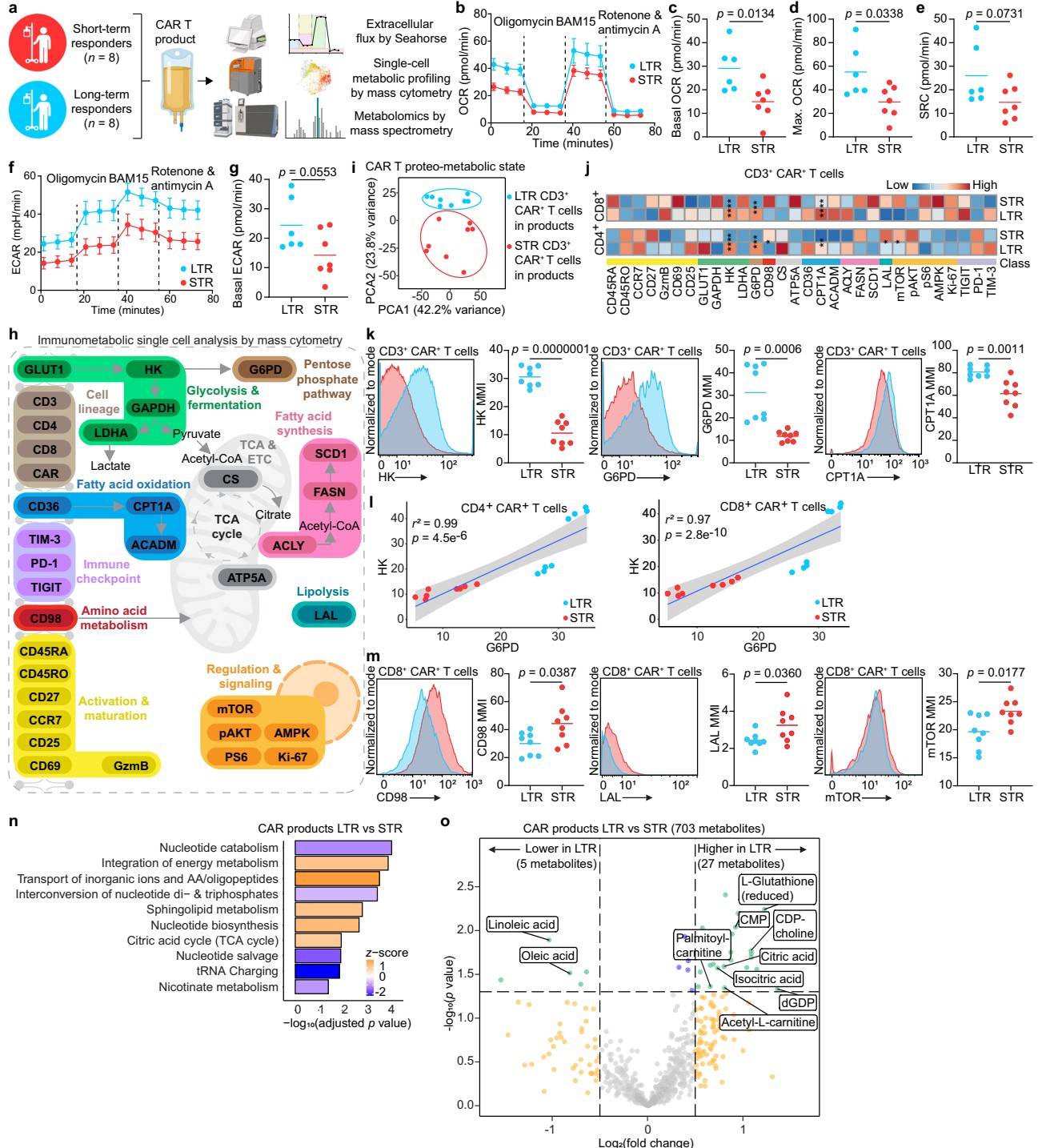

**Fig. 1 | CAR T cell products of long-term responders exhibit elevated oxidative phosphorylation, fatty acid oxidation, and pentose phosphate pathway activity.** Comparison of the immunometabolic features of pre-infusion CAR T cell products of short- and long-term responders. **a** Schematic of the experimental design (created in BioRender. Goldberg, L. (2026): https://BioRender.com/2lgakqd). **b** OCR plots (mean ± SEM) and quantification of basal respiration (**c**) maximal respiration (**d**) and spare respiratory capacity (**e**). **f** ECAR plots (mean ± SEM) and quantification of basal glycolysis (**g**). **h** Panel schematic of the immunologic proteins and metabolic pathways interrogated by mass cytometry (created using BioRender. Goldberg, L. (2026): https://BioRender.com/2lgakqd). **i** Metabolic state PCA of CD3 + CAR + T cells products, considering only the expression of metabolic proteins. **j** Heat map indicating z-score normalized median expression of

all CyTOF markers assessed in CD4+ and CD8 + CAR + T cells. **k** Representative histograms and analysis of statistically significant proteins in CD3 + CAR + T cells. **l** Linear correlation of median protein expression of HK and G6PD in CD3 + CAR + T cells, gray shading representing the 95% confidence interval. **m** Representative histograms and analysis of statistically significant proteins in CD8 + CAR + T cells. **n** Bar plot of statistically significant enriched metabolic pathways. **o** Volcano plot of differentially expressed abundant metabolites. MMI: median metal intensity. Data are mean from $n = 13$ (**b-g**), $n = 16$ (**i-o**) patients. Significance levels were calculated using two-tailed unpaired Student's $t$-tests. Significance values: *$p < 0.05$; **$p < 0.01$; ***$p < 0.001$; ****$p < 0.0001$. Source data are provided as a Source Data file.

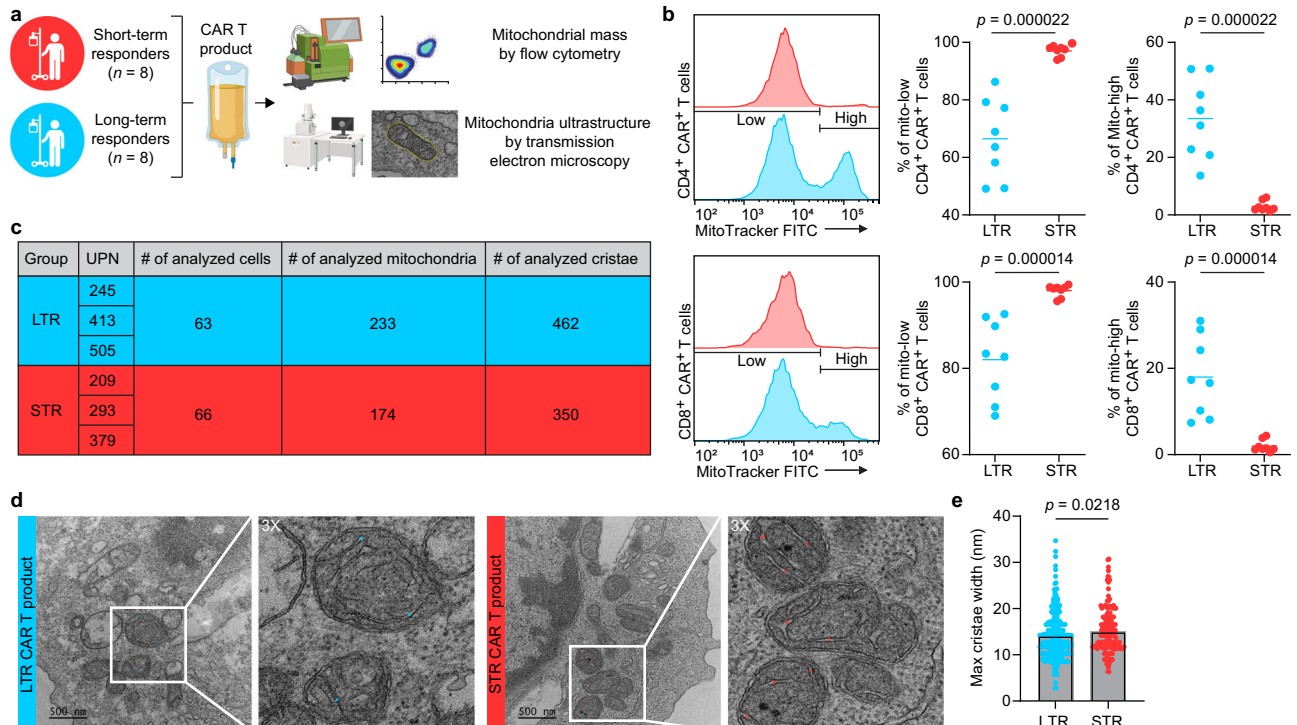

**Fig. 2 | CAR T products of long-term responders have increased mitochondrial mass and tighter cristae structure. a** Schematic of the experimental design (created in BioRender. Goldberg, L. (2026): https://BioRender.com/4r8x3vg). **b** Representative histograms and analysis of MitoTracker expression in CD4+ and CD8 + CAR + T cell products. **c** Table showing the number of ultrastructure components analyzed from transmission electron microscopy images. **d** Representative transmission electron microscopy images showing how mitochondrial cristae widths were measured. Scale bar, 500 nm. **e** Maximal cristae width assessed in CAR T cell products (462 cristae of LTR and 350 cristae of STR). Data are mean from $n = 16$ (**b**) and $n = 6$ (**e**) patients. Significance levels were calculated using two-tailed unpaired Student's $t$ tests (**b**) and two-tailed Mann–Whitney $U$-tests (**e**). Source data are provided as a Source Data file.

PPP activity (Fig. 1o, Supplementary Data 3c). Lastly, given the increased PPP activity observed in CAR T cells of LTR and the role of this metabolic pathway in maintaining cellular redox balance, we measured the reactive oxygen species levels. We found no difference between the two groups (Supplementary Fig. 3), supporting our previous metabolic assays showing that PPP activity mainly supports fatty acid oxidation, the TCA cycle, and nucleotide synthesis.

To further study the metabolic properties of the pre-infusion products, we measured mitochondrial mass and evaluated mitochondrial ultrastructure (Fig. 2a). The mitochondria of both CD4 and CD8 CAR T cells of LTR compared to STR had significantly higher mass (MitoTracker high), which is associated with superior metabolic fitness and antitumor activity[8] (Fig. 2b, Supplementary Fig. 4, Supplementary Fig. 5). A similar finding was observed in CD3 CAR T-negative cells, suggesting these mitochondrial changes are independent of CAR expression (Supplementary Fig. 2c). Moreover, measurement of the cristae width of hundreds of mitochondria (Fig. 2c-d) revealed that CAR T cell products of LTR vs. STR have significantly tighter mitochondrial cristae (Fig. 2e), which is associated with increased OXPHOS and antitumor activity[6]. Lastly, to further evaluate the mitochondrial features of the CAR T cells, we measured the mitochondrial membrane potential and found no differences between the two groups (Supplementary Fig. 6).

Since the metabolic profile of T cells is directly linked to differentiation[7], we compared the differentiation state of the pre-infusion products of the two sub-groups as measured by the abundance of various T cell subsets (stem cell, central memory, effector, effector memory, and TEMRA subsets[11]). We did not observe significant differences in the abundance of different T cell subsets measured using canonical markers CD45RA/CCR7 (Supplementary Fig. 7a-c), as well as the level of CCR7 in CD4+ and CD8 + CAR T cells (Fig. 1j),

suggesting that differences in CAR T cell metabolic functionality between the two sub-groups are not explained by their differentiation or memory markers. Overall, our multi-layer integrative analysis of pre-infusion CAR T cell products showed that products from LTR have increased OXPHOS, markers of fatty acid oxidation, and PPP activities as well as higher mitochondrial mass and tighter cristae, while products from STR have increased T cell activation.

## CAR T cells of LTR show high immunometabolic polyfunctionality post-infusion supported by the bone marrow (BM) microenvironment

To study the relationship between immunometabolism and CAR T cell anti-tumor activity in patients, we examined day 28 post-infusion BM samples (Fig. 3a), which allows us to investigate the microenvironment of both the leukemia as well as the post-infusion CAR T cells. Using our single-cell mass cytometry approach, which allows us to detect CAR T cells in the context of the broader cellular BM background, we compared changes in metabolic protein expression over time between the pre-infusion products and the day 28 BM samples of the two cohorts. We visualized our single-cell data by PCA considering only the expression of proteins associated with cellular metabolism and found that CAR T cells in the BM acquire distinct metabolic states post infusion (Fig. 3b). Similar to the pre-infusion products, CD3 CAR T cells in the BM at day 28 post-infusion of STR showed significantly higher expression of CD98 and ATP5a, which is consistent with effector activation[12]. However, CD3 CAR T cells of LTR exhibited significantly increased mTOR and pS6 expression compared to STR, suggesting a shift from a memory-to-effector metabolic phenotype of LTR CAR T cells from pre- to post-infusion, respectively (Fig. 3c). Indeed, longitudinal comparison of STR and LTR CAR T cells pre-infusion vs. post-infusion revealed that pre-infusion products of both cohorts had

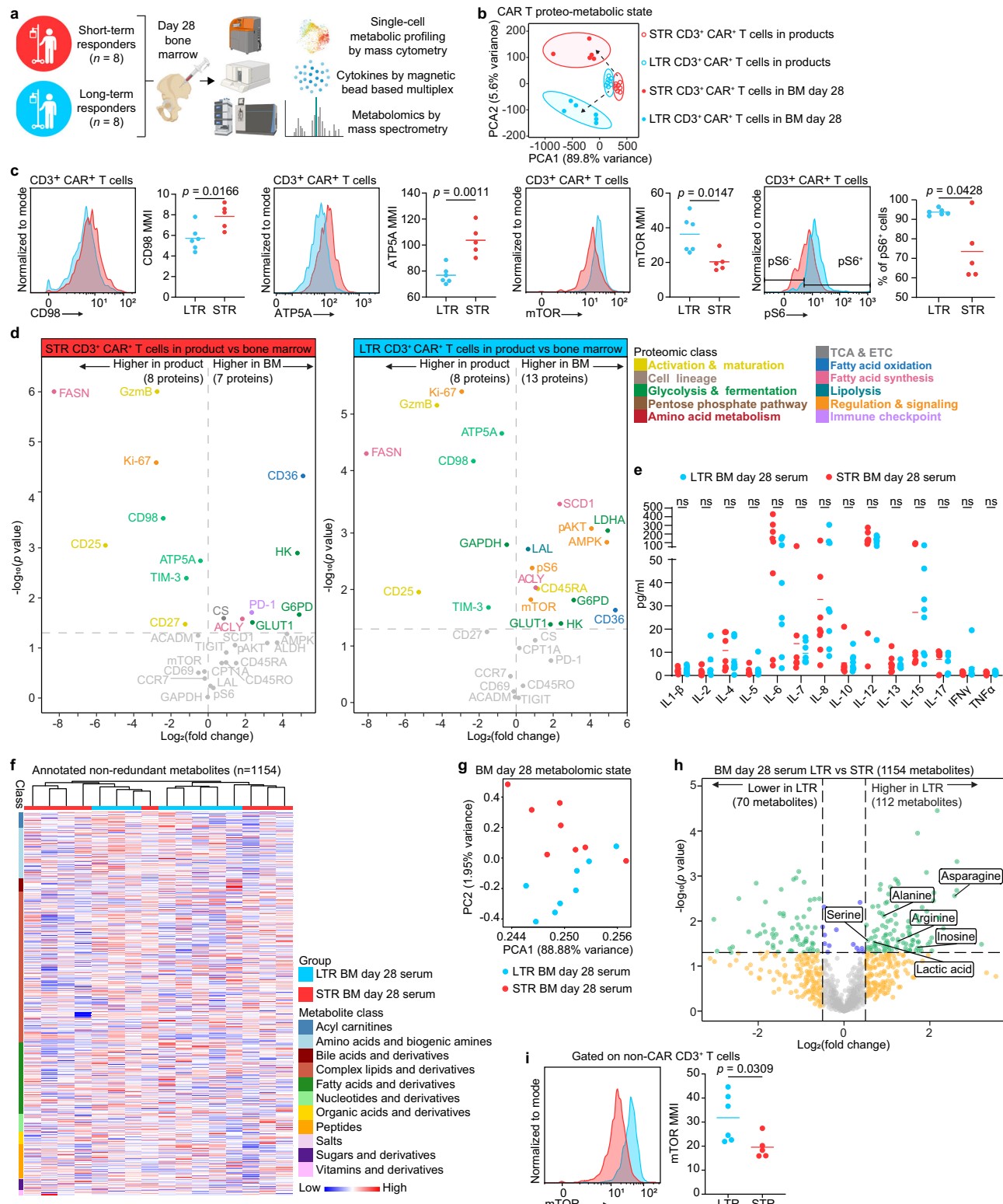

**Fig. 3 | CAR T cells in post-treatment bone marrow of long-term responders show increased mTOR signaling. a** Schematic of the experimental design (created in BioRender. Goldberg, L. (2026): https://BioRender.com/qdk05uc). **b** Metabolic state PCA of CD3 + CAR + T cell products and CD3 + CAR + T cells in the day 28 bone marrow, considering only the expression of metabolic proteins. **c** Representative histograms and analysis of statistically significant proteins in CD3 + CAR + T cells in day 28 bone marrow. **d** Volcano plots showing differentially expressed CyTOF markers in CD3 + CAR + T cell products versus CD3 + CAR + T cells in the day 28 bone marrow for both STR and LTR. **e** Day 28 bone marrow serum cytokine levels. **f** Unsupervised clustering based on patient outcome (LTR vs. STR) of nonredundant metabolites in day 28 bone marrow grouped by metabolite class. **g** Metabolomic state PCA of nonredundant metabolites in day 28 bone marrow. **h** Volcano plot of differentially expressed abundant metabolites in day 28 bone marrow of LTR vs. STR. Data are mean from $n = 11$ (bone marrow samples) and $n = 16$ (CAR products). **i** Representative histogram and analysis of mTOR expression in CD3+ non-CAR T cells. MMI: median metal intensity. Data are mean from $n = 11$ bone marrow samples (**c-d, i**), $n = 16$ (**b, e-h**) patients. ns, non-significant. Significance levels were calculated using two-tailed unpaired Student's $t$-tests. Source data are provided as a Source Data file.

higher levels of GzmB, CD25, CD98, ATP5A, FASN, Ki-67 and TIM-3 compared to CAR T cells in the BM at day 28. However, CAR T cells in the BM of LTR vs. STR upregulated more immunometabolic-relevant proteins post infusion (13 vs. 7), suggesting a higher capacity of immunometabolic plasticity (Fig. 3d). Specifically, CAR T cells in the day 28 BM of LTR had significantly higher expression of mTOR, pS6, pAKT (mTOR and pAKT pathways) and the metabolic sensor AMPK, as well as significant increases in the fermentative glycolysis enzyme LDHA, the endogenous fatty acid metabolism enzymes SCD1, LAL, and CD45RA, suggesting an effector immunometabolic phenotype. In contrast, CAR T cells in the day 28 BM of STR had significantly higher expression of PD-1 compared to the pre-infusion product, suggesting that CAR T cells of STR may have acquired an exhausted or over-activated phenotype post-infusion (Fig. 3d).

Given our longitudinal results showing distinct immunometabolic protein expression in CAR T cells post-infusion, we interrogated the cytokines and metabolites in the BM microenvironment of both cohorts. We focused our BM cytokine analysis on those related to T cell activation, proliferation, differentiation, and metabolism[13], including IL −1β, −2, −4, −5, −6, −7, −8, −10, −12, −13, −15, −17, IFNγ and TNFα, and found no significant differences between the two cohorts (Fig. 3e), suggesting that the differences in immunometabolism that we observed were not related to cytokines in the BM microenvironment. We then performed untargeted metabolomics and found 1154 non-redundant metabolites in the BM microenvironment (Fig. 3f). PCA analysis based on the metabolomic state revealed distinct clustering of the two cohorts, suggesting a distinct metabolic BM microenvironment in LTR compared to STR (Fig. 3g, Supplementary Data 4a-c). We found that LTR had significantly higher levels of three amino acids known to increase mTOR activity in T cells (asparagine, alanine, and arginine)[14], suggesting an extrinsic microenvironmental contribution to the increased mTOR activity seen in the CAR T cells of this cohort (Fig. 3h). Furthermore, we found that LTR had significant increases in serine and inosine, which are required for optimal T cell proliferation, enhanced adoptive immunotherapy, and can reverse CAR T cell exhaustion in preclinical models[15,16], as well as lactic acid, which preclinically improves adoptive immunotherapy[17] and might be related to increased LDHA expression seen in this group (Fig. 3h). To further explore the role of the BM microenvironment in promoting mTOR activity in LTR, we analyzed mTOR expression in the non-CAR T cell fraction in the BM, which is composed of non-transduced T cells from the CAR T cell product as well as endogenous reconstituting T cells. We found significantly increased mTOR activity in LTR compared to STR (Fig. 3i), further supporting a crosstalk between BM nutrients and mTOR immunometabolic activity of T cells in the BM. Overall, we found that CAR T cells in post-infusion BM samples from LTR have high levels of mTOR signaling, high metabolic polyfunctionality, and an effector immunometabolic profile, and that the BM microenvironmental milieu of LTR may promote mTOR activity for improved anti-tumor activity.

### Inhibiting mTOR signaling in CAR T cells alters immunometabolism and enhances function

Intrigued by our pre- and post-infusion results suggesting a significant role for mTOR activity on the efficacy of CAR T cells of LTR, we asked whether we could harness the mTOR pathway to enhance antitumor activity of CAR T cells of STR. To that end, we manufactured CAR T cells from STR-derived leukapheresis products with or without the mTOR inhibitor rapamycin to transiently inhibit mTOR activity with the goal of isolating the effects of mTOR signaling in the STR-derived products (Fig. 4a). Rapamycin-treated and untreated CAR T cell products had similar in vitro expansion (Fig. 4b) and CD4 and CD8 abundances (Fig. 4c). Intrigued by the similarity in the expansion of both groups and given the immunosuppressive function of rapamycin by inhibiting entry into the cell cycle, we measured the cell cycle

distributions of both groups. We observed that on day 11, most CAR T cells in both groups exited the non-dividing G0 stage and found slight significant differences between the two groups in the S and G0/G1 phases (Fig. 4d-e), suggesting that starting to add rapamycin on day 4 post-anti-CD3/CD28 beads activation did not prevent activated CAR T cells from dividing. In line with the role of mTOR as a master metabolic regulator, rapamycin-treated enriched CAR T cells showed significant decrease in basal respiration (Fig. 4f-i) and glycolysis (Fig. 4j-k) yet exhibited higher spare respiratory capacity (SRC) (Fig. 4i), which is associated with memory development and enhanced anti-tumor activity[6,7], compared to control enriched CAR T cells. Using RNA gene set enrichment analysis (Fig. 4l, Supplementary Data 5a), we confirmed that rapamycin inhibited mTOR-MYC signaling/targets as well as downregulated all major metabolic pathways (OXPHOS, glycolysis and fatty acid metabolism) and T cell activation/effector pathways (IL-2-STAT5, IL-6-JAK-STAT3, allogenic activity, and interferon response). Moreover, rapamycin upregulated Wnt/β-catenin signaling, including statistically significant upregulation of the T cell memory genes *TCF7, LEF1, FOXO1* in rapamycin-treated products (Supplementary Data 5b). Furthermore, effector genes *GZMB, GNLY, GZMA, PRF1, IL-2, IFNγ* as well as the master regulator of exhaustion *TOX* were significantly downregulated in rapamycin-treated CAR T products (Fig. 4m). Of note, consistent with post-translational mTOR inhibition by rapamycin, *MYC* was significantly upregulated, yet *MYC* targets were significantly downregulated (Fig. 4l-m).

We confirmed our RNA results at the single cell proteomic level and found significant downregulation of mTOR-pS6 and the metabolic proteins HK, GAPDH, G6PD, CD98, CS, ATP5A, FASN in rapamycin-treated products (Fig. 4n). Moreover, rapamycin treatment resulted in significant downregulation of both CD69 and GzmB, which is consistent with less activated and effector T cell state (Fig. 4n). We then performed untargeted intracellular metabolomics and identified 560 nonredundant metabolites; pathway enrichment analysis revealed enriched features associated with amino acid, sphingolipid, and glycerophospholipid metabolism in CAR T cell products following rapamycin treatment (Fig. 4o, Supplementary 6a-c). Consistent with enriched glycerophospholipid and amino acid metabolism, levels of the phospholipids phosphoethanolamine, glycerol 3-phosphate, CDP-ethanolamine as well as the amino acids glutamine, serine, and cis-4-Hydroxy-D-proline were increased in rapamycin-treated CAR T cell products (Fig. 4p, Supplementary Data 6c). Moreover, levels of sedoheptulose 7-phosphate and glyceraldehyde 3-phosphate were decreased following rapamycin treatment, suggesting decreased non-oxidative PPP metabolism (Fig. 4p, Supplementary Data 6c). When we evaluated mitochondrial features of CAR T cells treated with or without rapamycin, we found that those treated with rapamycin had significantly higher mitochondrial mass (Fig. 4q) and significantly lower mitochondrial membrane potential (Fig. 4r), both of which are associated with memory T cell precursors and superior antitumor activity[18]. Next, we measured the cristae width of hundreds of mitochondria and determined that rapamycin-treated CAR T cell products had significantly looser mitochondrial cristae, which is consistent with their less-activated metabolic state (Fig. 4s-t). Lastly, similar to pre-infusion products, we did not observe significant differences in the abundance of different T cell subsets in CAR T cells from the day 28 bone marrow (Supplementary Fig. 7d). Overall, by transiently inhibiting mTOR with rapamycin, we induced a metabolic and activation rest of the CAR T cells.

To determine whether STR-derived CAR T cells treated with rapamycin had better anti-tumor efficacy in vivo compared to untreated CAR T cells, we inoculated NSG mice with Nalm6 leukemia cells and treated mice with CAR T cells (±rapamycin) five days later (Fig. 5a). We observed significantly enhanced tumor control reflected by longitudinal bioluminescence imaging (Fig. 5b-c) and prolonged survival in mice injected with rapamycin-treated CAR T cell products

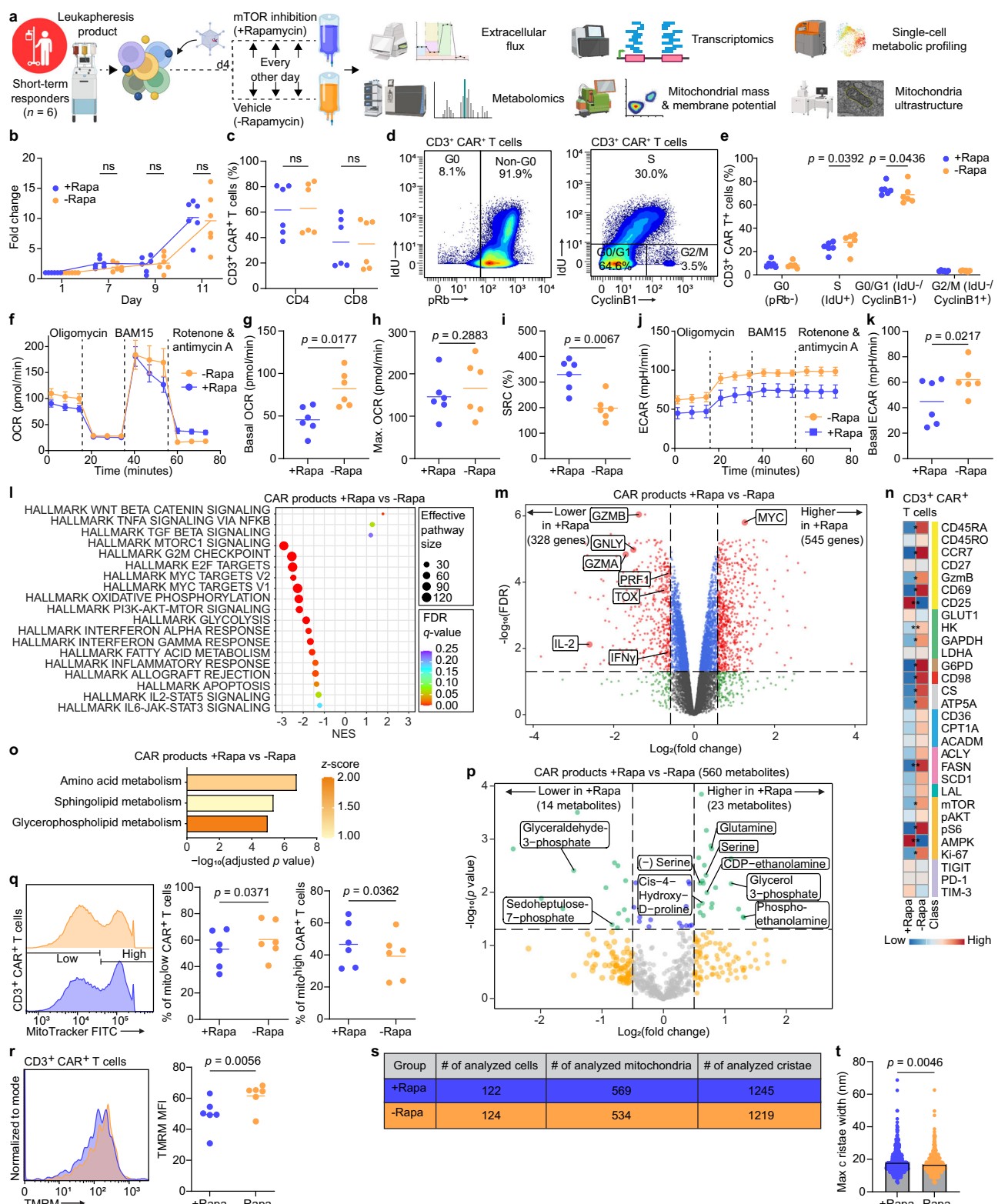

(Fig. 5d). Furthermore, rapamycin-treated CAR T cells had significantly higher in vivo expansion, with absolute CAR+ cell counts in the peripheral blood of mice reaching a mean of tenfold higher compared with untreated CAR T cells on day 14 post injection (Fig. 5e). Lastly, we asked whether our rapamycin strategy would also enhance the in vivo efficacy of CAR T cells from LTR. To this end, we repeated our in vivo study using LTR-derived CAR T cells manufactured with or without rapamycin (Fig. 5f). Consistent with our observations for STR-derived

CAR T cells, we observed significant reduction in tumor burden (Fig. 5g-h) and prolonged survival in mice injected with rapamycin-treated CAR T cell products from LTR (Fig. 5i). Similar results were observed in a central nervous system (CNS) B-ALL model, where NSG mice were engrafted with O18z leukemic cells, derived from a child with suspected CNS involvement leukemia[19] (Fig. 6a, d). We observed a significant reduction in tumor burden and prolonged survival in mice injected with rapamycin-treated CAR T cell products compared with

**Fig. 4 | Transient mTOR inhibition induces metabolic and activation rest and improves metabolic fitness of CAR T cell products derived from short-term responders.** Comparison of CAR T cells products manufactured with ( + Rapa) or without (-Rapa) rapamycin. **a** Schematic of the experimental design (created in BioRender. Goldberg, L. (2026): https://BioRender.com/c59o90s). **b** Mean fold expansion. **c** Abundance of CD4+ and CD8 + CAR T cells. **d** Representative biaxial plots to measure cell cycle phases in CD3 + CAR + T cells. **e** Percentage of CD3 + CAR + T cells in each cell cycle phase as measured by mass cytometry. **f** OCR plots (mean ± SEM) and quantification of basal respiration (**g**) maximal respiration (**h**) and spare respiratory capacity (**i**). **j** ECAR plots (mean ± SEM) and quantification of basal glycolysis (**k**). Gene set enrichment analysis (**l**) and volcano plot of differentially expressed genes (**m**) in CAR T cells with or without rapamycin. **n** Heat map indicating z-score normalized median expression of all CyTOF markers assessed in CD3 + CAR T cells. **o** Bar plot of statistically significant enriched metabolic pathways in CAR T cells treated with rapamycin. **p** Volcano plot of differentially expressed abundant metabolites in CAR T cell products treated with or without rapamycin. **q** Representative histograms and analysis of MitoTracker expression in CD3 + CAR + T cell. **r** Representative histograms and analysis of TMRM expression in CD3 + CAR + T cell. **s** Table showing the number of ultrastructure components analyzed from transmission electron microscopy images. **t** Maximal cristae width assessed in CAR T cell products (1245 cristae of +Rapa and 1219 cristae of -Rapa). Scale bar, 500 nm. Data are mean from n = 6 patient-derived products (**b-t**). Significance levels were calculated using two-tailed paired Student's *t* tests (**b–r**) and two-tailed Mann–Whitney *U*-tests (**t**). Significance values: ns, non-significant; *$p < 0.05$; **$p < 0.01$. Source data are provided as a Source Data file.

untreated CAR T cells from both STR (Fig. 6b-c) and LTR (Fig. 6e-f). Lastly, after showing enhanced anti-leukemic activity in both STR and LTR patient-derived CAR T cells manufactured with rapamycin, in two independent models of leukemia (Nalm6 and 018z), we were intrigued by the possible difference in overall survival of STR with rapamycin compared to the LTR with rapamycin in the Nalm6 model (Fig. 5d, i), with the caveat that a direct comparison of mice treated with STR and LTR is limited due to product/donor variability. Based on our findings of decreased CAR T cell activation induced by rapamycin (Fig. 4) and the relationship between precise patient-specific control of activation level and CAR T cell functionality[20], we hypothesized that rapamycin may have a greater effect on decreasing the activation of STR compared to LTR CAR T cells. STR and LTR CAR T cells without rapamycin had similar levels of CD69 (Fig. 1j). However, we found lower expression of CD69 in STR with rapamycin compared to LTR with rapamycin, including in the patient-derived CAR T cells used in the Nalm6 mice experiments (Supplementary Fig. 8), suggesting that rapamycin decreased the activation of STR CAR T cells to a greater extent than LTR CAR T cells and may explain the difference in mouse survival in the Nalm6 model. Overall, our data suggest that upfront incorporation of the mTOR inhibitor rapamycin in CAR T cell manufacturing may be a means to enhance anti-tumor efficacy in patients.

## Discussion

As of early 2024, more than 34,000 patients in the United States have received autologous CAR T cells, mainly targeting CD19 for hematological malignancies[21]. Although the field of cellular therapy has gained substantial clinical insights into optimizing CAR T cell therapy, the immunometabolic intricacies of CD19-CAR T cells in patients pre- and post-infusion have not been fully elucidated. Here we applied an integrative, multi-layer approach to investigate the immunometabolic determinants of CAR T cell response in patients with B-ALL. First, we revealed that pre-infusion CAR T cell products from LTR have an altered immunometabolic signature acquiring features of memory T cells, which is associated with improved anti-tumor efficacy and persistence, including increased OXPHOS, markers of fatty acid oxidation, and PPP activities (Fig. 1) as well as higher mitochondrial mass and tighter cristae (Fig. 2) compared to products from STR[5–8,22]. These metabolic differences were not mirrored by changes in surface phenotypic markers for T cell subset and differentiation state (Supplementary Fig. 7a-c), emphasizing the distinct biology of CAR T cells compared to conventional T cells and the importance of in-depth metabolic functional and phenotypic assays while studying CAR T cells. Moreover, our data complements previous observations showing that pre-infusion CD19-CAR T cell products of patients in complete response compared to non- or partial responders have higher mitochondrial volume by 3D confocal microscopy[23], and that mitochondrial transfer enhances CD19-CAR T cell metabolic fitness and antitumor efficacy in a preclinical model[8].

Second, we found that in post-infusion BM samples, CAR T cells of LTR have increased mTOR-pS6, pAKT, AMPK and LDHA expression,

supporting metabolic plasticity, and effector immunometabolic profile compared to those of STR as well as a microenvironmental milieu promoting mTOR activity (Fig. 3). These findings align with previous evidence of CAR T cells from non-responders or early relapsed patients stimulated with CD19 that displayed transcriptional programs associated with dysfunctional mTOR signaling and lower metabolic activity[24]. Furthermore, actively cycling CAR T cells isolated from a patient with a decade-long leukemia remission showed upregulation in *LDHA* and OXPHOS pathways, suggesting that both aerobic glycolysis and mitochondrial respiration are increased in CAR T cells that persist long-term[25]. Finally, our findings are consistent with the significantly increased *AMPK* expression in pre-infusion CD19-CAR T cell products of patients in complete response compared to non- or partial responders as well as decreased functionality of CD19-CAR T cells upon *AMPK* inhibition[23]. Overall, our findings suggest that both intrinsic and extrinsic factors contribute to CAR T cell functionality post-infusion, including a cross-talk between microenvironmental nutrients and the CAR T cell immunometabolic state in patients.

Lastly, intrigued by our observations related to mTOR, we harnessed the mTOR pathway with the mTOR inhibitor rapamycin to alter the immunometabolism and enhance anti-tumor activity of patient-derived CAR T cells (Figs. 4–6). Mechanistically, by inhibiting mTOR activity, we induced transient metabolic and activation rest of CAR T cell products. This rest was associated with increased SRC and mitochondrial mass and decreased mitochondrial membrane potential, fundamental metabolic features associated with memory T cell precursors and superior antitumor activity[6,7,18]. Indeed, in a leukemia mouse model, patient-derived mTOR inhibited CAR T cell products showed improved expansion post-infusion (Fig. 5e), which resulted in lower tumor burden and longer survival compared to mice treated with control CAR T cells (Figs. 5–6). Our findings support the idea of signaling rest as a means to improve T cell functionality. Indeed, both CAR T cells and tumor-infiltrating lymphocytes subjected to a signaling rest via BRAF/MEK[26] and tyrosine kinase inhibitors[27], respectively, underwent reprogramming associated with improved in vivo expansion. Based on our studies, we hypothesize that transient treatment with rapamycin, a clinically available drug, offers a simple and rapid metabolic approach to improve current CAR T cell manufacturing practices.

Our study is limited by the number of patients in our cohort (n = 16), which was restricted by the scarcity of patient clinical samples, especially BM samples that require an invasive procedure to procure. We also were careful in our patient selection, pairwise matching the sub-groups to mitigate potential confounding variables, which further limited our number of patients. Furthermore, our single-cell mass cytometry approach was limited to ~40 proteins, capping the number of metabolic pathways we could interrogate simultaneously. Finally, our findings are based on a manufacturing approach that depletes leukapheresis products of CD14+ monocytes and CD25+ Tregs, followed by positive selection of CD62L + T cells, and should not be generalized to other manufacturing approaches without further

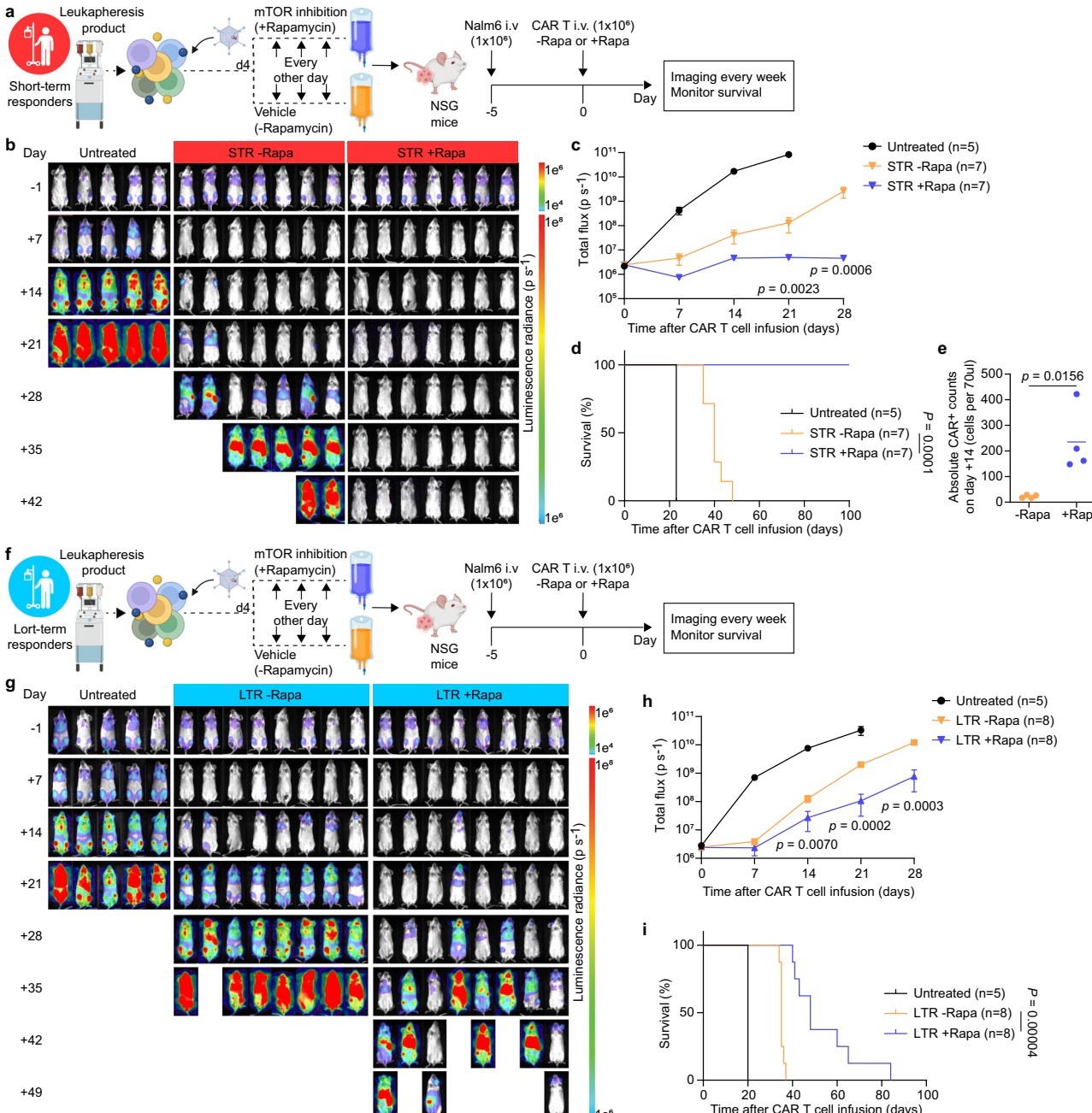

**Fig. 5 | Patient-derived CAR T cells demonstrate enhanced antitumor activity following transient mTOR inhibition. a** Experimental setup of the in vivo leukemia model testing short-term responder-derived CAR T cell products treated with or without rapamycin (created in BioRender. Goldberg, L. (2026): https://BioRender.com/y346vyd). NSG mice were intravenously injected with $1 \times 10^6$ Nalm6 cells. Five days later, mice were randomly assigned to three groups and treatment groups were infused intravenously with $1 \times 10^6$ CD3 + CAR + T cells. **b–c,** Weekly bioluminescence imaging analysis monitoring tumor burden in the different groups. Untreated (*n* = 5 mice); STR (*n* = 7 mice per treatment group). **d** Kaplan−Meyer analysis of mouse survival (*n* = 7 mice per treatment group, representative of two independent experiments/patients). **e** Peripheral blood collected on day 14 post-CAR T cell treatment was analyzed for presence of CAR T cells by flow cytometry (*n* = 4 mice per treatment group) **f** Experimental setup of the in vivo leukemia model testing long-term survivor-derived CAR T cell products treated with or without rapamycin (created in BioRender. Goldberg, L. (2026): https://BioRender.com/y346vyd). **g−h,** Weekly bioluminescence imaging analysis monitoring tumor burden in the different groups. Untreated (*n* = 5 mice); LTR (*n* = 8 mice per treatment group). **i** Kaplan−Meyer analysis of mouse survival (*n* = 8 mice per treatment group). Data are mean ± SEM (**c, h**) and mean (**e**). Significance levels were calculated using two-tailed unpaired Mann−Whitney *U* tests (**c, h**), two-tailed paired Student's *t* tests (**e**) or log-rank Mantel−Cox tests (**d, i**). Source data are provided as a Source Data file.

experimental support. Nevertheless, our multi-layer integrative approach allowed us to decipher some of the fundamental immuno-metabolic determinants of CAR T cell response in leukemia patients receiving CD19-CAR T cell therapy. We believe our work paves the way for a larger validation cohort including novel single-cell proteomic approaches being currently developed[28].

## Methods

All studies were conducted in accordance with protocols approved by the City of Hope's Internal Review Board (IRB) and Institutional Animal Care and Use Committee (IACUC). All relevant animal use guidelines and ethical regulations were followed. Further information on research design is available in the Reporting Summary linked to this article.

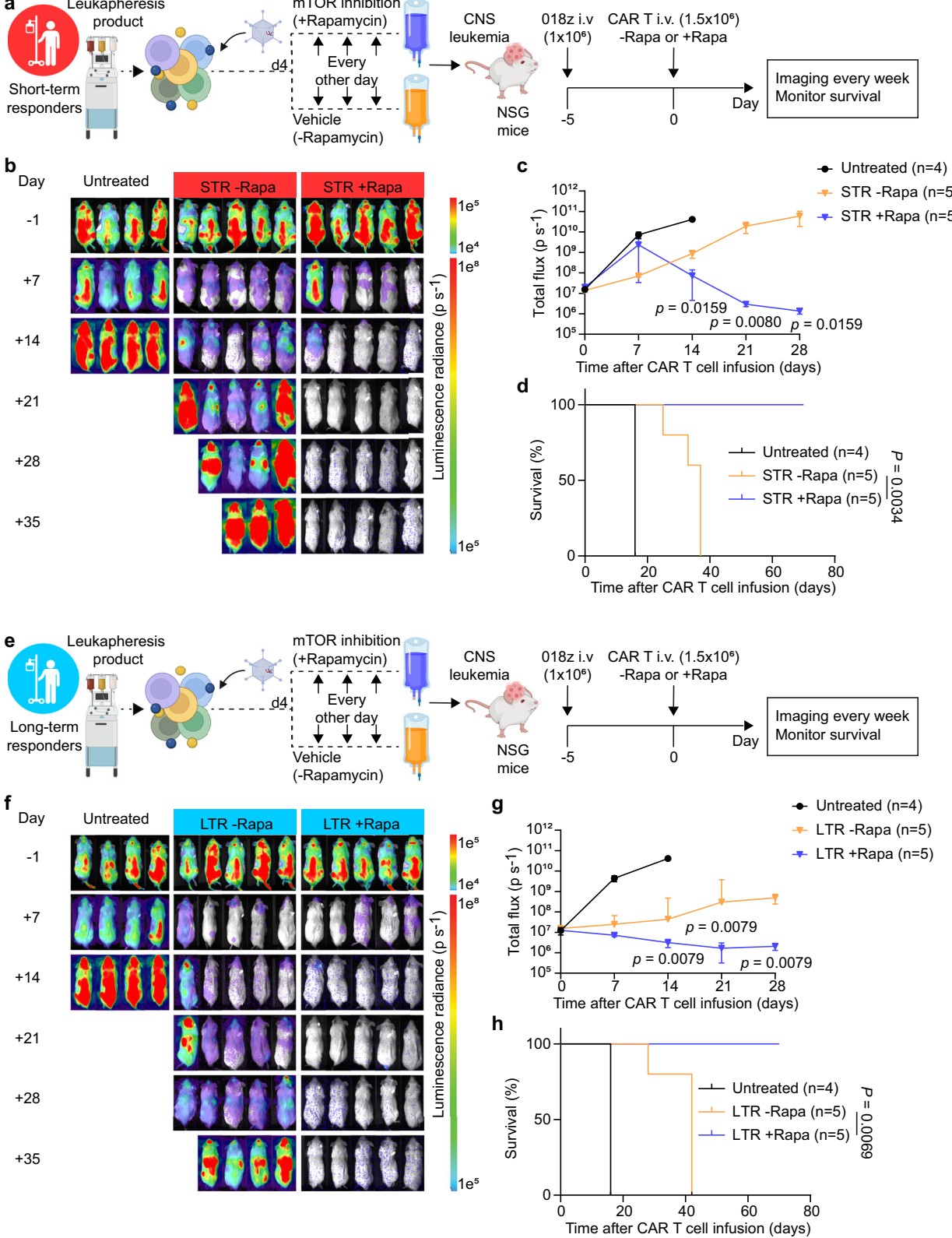

## Patient samples and CAR T cell therapy

De-identified clinical samples were collected from a subset of adult patients with relapsed/refractory B-ALL who received investigational CD19-specific CAR T cells in a phase 1/2 clinical trial (NCT02146924) at our institution[29]. The study was conducted in accordance with the Declaration of Helsinki and with the approval of the City of Hope Internal Review Board (IRB #13447). Written informed consent was obtained from all participating patients. CAR T cells were manufactured from isolated T naïve/memory (Tn/mem) cells as previously described[3]. Autologous CD19R(EQ)28ζ/EGFRt+ Tn/mem cell products were administered by intravenous infusion in a single dose following a 3-day lymphodepleting regimen.

**Fig. 6 | Patient-derived CAR T cells demonstrate enhanced antitumor activity following transient mTOR inhibition in a CNS leukemia model. a** Experimental setup of the in vivo CNS leukemia model testing short-term responder-derived CAR T cell products treated with or without rapamycin (created in BioRender. Goldberg, L. (2026): https://BioRender.com/aywhx3w). NSG mice were intravenously injected with $1 \times 10^6$ 018z cells. Five days later, mice were randomly assigned to three groups and treatment groups were infused intravenously with $1.5 \times 10^6$ CD3 + CAR + T cells. **b**, **c** Weekly bioluminescence imaging analysis monitoring tumor burden in the different groups. Untreated ($n$ = 4 mice); STR ($n$ = 5 mice per treatment group). **d** Kaplan−Meyer analysis of mouse survival ($n$ = 5 mice per treatment group).

**e** Experimental setup of the in vivo CNS leukemia model testing long-term responder-derived CAR T cell products treated with or without rapamycin (created in BioRender. Goldberg, L. (2026): https://BioRender.com/aywhx3w). **f**, **g** Weekly bioluminescence imaging analysis monitoring tumor burden in the different groups. Untreated ($n$ = 4 mice); LTR ($n$ = 5 mice per treatment group). **h** Kaplan−Meyer analysis of mouse survival ($n$ = 5 mice per treatment group). Data are mean ± SEM (**c**, **g**). Significance levels were calculated using two-tailed unpaired Mann−Whitney $U$-tests (**c**, **g**) or log-rank Mantel−Cox tests (**d**, **h**). Source data are provided as a Source Data file.

## Metabolic analysis using the Seahorse XF96 extracellular flux analyzer

Mitochondrial oxygen consumption rate (OCR) and extracellular acidification rate (ECAR) were measured using the Seahorse Bioscience XF96 Extracellular Flux Analyzer (Agilent). CAR T cell products were enriched for CAR T cells using the EasySep™ Human EGFR Positive Selection Kit (StemCell Technologies, 100-1131) according to the manufacturer's instructions, and 150,000 CAR T cells per well (minimum of two wells per sample) were seeded in Seahorse XFe96 PDL cell culture microplates (Agilent, 103799-100). Cells were suspended in Seahorse XF RPMI medium (200 ul/well, Agilent, 103576-100) supplemented with 10 mM glucose, 2 mM glutamine, and 1 mM sodium pyruvate. OCR was measured following sequential injections of 2.5 μM oligomycin A, 2.5 μM BAM15, and 0.5 μM rotenone/antimycin A (all Agilent, Seahorse XF T Cell Metabolic Profiling Kit, 103772-100) according to the manufacturer's instructions.

## CyTOF

Metal-isotope conjugated monoclonal antibodies (Supplementary Data 2) targeting T cell phenotypic, activation, and metabolic proteins were used to evaluate the metabolic and activation properties of patient-derived CAR T cells in vitro and ex vivo. Non-commercially available metal-tagged antibodies were generated by conjugating each purified antibody to its respective metal isotope following standard procedures (Maxpar Antibody Labeling User Guide PRD002). Samples were stained according to the manufacturer's instructions (Maxpar® Phospho-Protein Staining PRD016) and acquired on a Helios mass cytometer (Standard Biotools). CyTOF data were normalized using bead-based time normalization according to manufacturer's instructions, then cleaned using the Gaussian parameter discrimination method (Supplementary Fig. 1) and manually debarcoded in OMIQ according to the barcode key for each sample set. Cleaned and debarcoded.fcs files were then reuploaded to OMIQ as live, intact single cells and manually gated to resolve CD8+ and CD4 + , CAR+ and CAR- T cells (only non-CAR T cells in leukapheresis samples) for further analysis.

## Metabolomics

Untargeted metabolomics was performed on CAR T cells and BM of LTR and STR as previously described[4,30]. Briefly, metabolites from $1.0$-$3.0 \times 10^6$ CAR T cells were extracted using 0.4 mL of solvent mixture (water: methanol: acetonitrile, 1:2.25:0.75, v/v/v) and 3 freeze-thaw (30 s each) cycles. For BM, 3-fold excess acetonitrile: methanol (3:1) was added for protein removal. The resultant metabolite extracts from CAR T cells and BM were recovered by centrifuging at $13{,}226 \times g$ at 4 °C for 15 min. Samples were pooled for quality control purposes (Pool QC) and injected at every five samples to monitor instrument performance and correct instrument drift. The LC-MS/MS was performed on both polar and hydrophobic metabolites using an Ultimate 3000 RSLC nano coupled to an HPG pump or a Vanquish Flex UPLCs coupled to Orbitrap Fusion Lumos Tribrid mass spectrometer (Thermo Fisher Scientific) in HILIC LC-MS and RP-LC-MS, operated in positive and negative ionization modes as described previously[30]. Raw data were analyzed on Compound Discoverer 3.2 using HMDB, KEGG, mzCloud

and LipidMaps databases. Minimum peak intensities of 70,000 and 100,000 were used as thresholds for negative and positive mode data, respectively. SERRF was used for normalization[31]. Compound annotations were manually verified. Only compounds observed in the Pool QC with an area <25% RSD, a mass error under ±5 ppm mass error and a < 0.2 min retention time drift were selected for downstream data analysis. The data were further normalized using the vsn package in R (v4.4.0). A Welch's t-test was used to identify metabolic differences between STR and LTR. Only metabolites with $p < 0.05$ were considered statistically significant. Redundancy within and across chromatographies for significantly different metabolites was addressed by selecting the metabolite with the highest area in the Pool QC samples. Only endogenous metabolites were used for data presentation and pathway analysis using Ingenuity Pathway Analysis (QIAGEN).

## Antibodies and flow cytometry

For surface staining, cells were incubated with fluorochrome-conjugated monoclonal antibodies (mAbs) to CD3 (BD Bioscience, 563109, 347347), CD4 (BD Bioscience, 347324), CD8 (BD Biosciences, 348793), EGFR (Biolegend, 352906, 352910). Cells were resuspended in FACS buffer, which consists of HBSS (Gibco, 14175095), 2% FBS and NaN3 (Sigma, S8032), and incubated with antibodies at 4 °C in the dark. For mitochondrial mass staining, cells were stained with MitoTracker Green (Invitrogen, M7514) at 200 nM per the manufacturer's protocol in addition to the surface antibody cocktail. For mitochondrial membrane potential, cells were stained with Tetramethylrhodamine, Methyl Ester, Perchlorate (TMRM) (Invitrogen, T668) at 100 nM per the manufacturer's protocol. To determine the level of ROS, the cells were incubated with 5 μM of CellROX Deep Red (Invitrogen, C10422) in PBS for 30 min at 37 °C per the manufacturer's protocol. Following a washing step with FACS buffer and surface antibody staining, DAPI (Invitrogen, D21490) was added for viability staining before analysis. Flow cytometry was performed using MACSQuant Analyzer 10 (Miltenyi Biotech, 130-096-343), according to the manufacturer's protocol. All flow cytometry data were compensated and gated to resolve live, intact, single lymphocytes using FlowJo (FlowJo™ Software, Version 10.8.1. Ashland, OR: Becton, Dickinson and Company; 2024.). The mean fluorescence intensity (MFI) and relevant percent abundance were exported from FlowJo for downstream analysis.

## Transmission electron microscopy

CAR T cell products ($1 \times 10^6$ cells) were fixed with 2.5% glutaraldehyde, 0.1 M Cacodylate buffer (Na(CH3)2AsO2 ·3H2O), pH7.2 at 4 °C. Standard sample preparation for TEM was followed, including post-fixation with osmium tetroxide, serial dehydration with ethanol, and embedment in Eponate[32] 70-nm-thin sections were acquired by ultramicrotomy, post-stained, and examined on an FEI Tecnai 12 transmission electron microscope equipped with a Gatan OneView CMOS camera.

Mitochondrial cristae width was manually measured using ImageJ software (version 1.53)[33] 704 pixels were set as 500 nm based on the image scale bar, and max cristae width per mitochondrion was used to compare differences between groups. Measurements of max cristae width were performed in a blinded fashion to prevent biased outcomes.

## Cytokine analysis

Bone marrow serum samples were analyzed for 30 cytokines using the Human Cytokine Thirty-Plex Antibody Magnetic Bead Kit (Invitrogen, LHC6003M) per the manufacturer's protocol. Briefly, Invitrogen's multiplex bead solutions were vortexed for 30 s and 25ul was added to each well and washed twice with Wash Buffer. Bone marrow serum samples were diluted 1:2 with assay diluent and loaded onto a Greiner flat-bottom 96-well microplate containing incubation buffer (50 ul/ well). Cytokine standards were reconstituted with assay diluent, and serial dilutions were prepared in parallel and added to the plate. Samples were incubated at room temperature for 2 h on a plate shaker at 500 rpm in the dark. After washing the plate three times with wash buffer (200 ul/well/wash) using a magnetic device designed to accommodate a microplate, biotinylated Detector Antibody was added (100 ul/well). The plate was incubated on a plate shaker for 1 h, washed three times as above, and streptavidin-phycoerythrin was added (100ul/ well). The plate was incubated on a plate shaker for 30 min and washed as above. After the final wash, each well was resuspended in 150 ul wash buffer and shaken for 1 min. The assay plate was then transferred to the Flexmap 3D Luminex system (Luminex Corp.) for analysis. Cytokine concentrations were calculated using Bio-Plex Manager 6.2 software with a five-parameter curve-fitting algorithm applied for standard curve calculations for duplicate samples.

## Generation of CAR T cells with rapamycin

Tn/mem cells were selected from leukapheresis products, activated overnight with GMP Human T-expander CD3/CD28 Dynabeads™ (Dynal Biotech Cat#11141D) at a 1:3 ratio (T cell:bead), and transduced at MOI of 1 with CD19R(EQ):CD28:ζ/EGFRt clinical construct in RPMI 1640 medium with 10% FBS, 5 µg/ml protamine sulfate (Fresenius Kabi, 22905), 50 U/mL rhIL-2 (Novartis Pharmaceuticals, NDC0078-0495-61), and 0.5 ng/mL rhIL-15 (CellGenix, 1013-050). Four days after bead stimulation, CAR T cells were treated with rapamycin (50 nmol/L) (LC Laboratories, R-5000) or vehicle (DMSO). Media containing rapamycin or vehicle was refreshed every other day throughout manufacturing. Six days after activation, CD3/CD28 Dynabeads™ beads were removed using a DynaMag™-5 Magnet (Invitrogen, 12303D). Cells were maintained at $1 \times 10^6$ cells/mL in RPMI 1640 media,10% FBS, 25 U/mL rhIL-2, and 0.25 ng/mL rhIL-15.

## RNA extraction, sequencing, and analysis

RNA from 100,000 cells from each CAR T cell product was extracted with the miRNeasy Mini Kit (QIAGEN, 217004), and RNA quality was assessed using the Agilent 2100 Bioanalyzer according to the manufacturer's instructions. Sequence alignment and gene counts: RNA-Seq reads were trimmed to remove sequencing adapters using Trimmomatic[34] and polyA tails using FASTP[35]. The processed reads were mapped back to the human genome (hg38) using STAR software (v. 2.6.0.a)[36]. The HTSeq software (v.0.11.1)[37] was applied to generate the count matrix, with default parameters. Differential expression analysis was performed by adjusting raw read counts to normalized expression values using the TMM method implemented in edgeR[38]. Briefly, for between-treatment comparisons, general linear models (GLMs) were used to identify differentially expressed genes (DEGs) between the +Rapa and -Rapa conditions, with TMM-normalized expression levels as the dependent variable. Genes with an FDR-adjusted $p$-value less than 0.05 and with a fold change (FC) greater than 2 or less than 0.5 were considered as significant up- and down-regulated genes, respectively. Pathway analysis was conducted using GSEA algorithm implemented in clusterProfiler (v.4.14.3) package in R[39–41], where a ranked list of whole genes according to their log2 fold change and $p$-values are provided.

## Mouse xenograft studies

Immunocompromised NOD scid IL2Rgamma$^{null}$ (NSG) mice were purchased from The Jackson Laboratory and maintained at the Animal Resource Center of City of Hope. All mouse experiments were performed in accordance with protocols approved by the Institutional Animal Care and Use Committee (IACUC: 21034). NSG mice (6–10 weeks old; equal number female and male) received $1.0 \times 10^6$ Nalm6 (ATCC, CRL-3273) or $1.0 \times 10^6$ 018z leukemia cells by intravenous injection to establish tumors. After 5 days, mice were randomized into groups to ensure equal tumor burden and treated with PBS (untreated), $1.0 \times 10^6$ (Nalm6 model) or $1.5 \times 10^6$ (018z model) CD19 CAR T cells manufactured with rapamycin or vehicle via intravenous tail injection. Tumor burden was monitored weekly by live mouse imaging using the LagoX optical imaging system (Spectral Instruments Imaging). For imaging, mice were given XenoLight D-luciferin potassium salt (Perkin Elmer, 122799) by intraperitoneal injection. Aura Imaging Software (Spectral Instruments Imaging) was used to analyze imaging. Mouse experiments continued until IACUC guidelines recommended euthanasia (pain, distress, or >20% weight loss compared to control or age-matched mice).

## Visualization

Schematic representations were created with BioRender.com. Figures were prepared in Illustrator (Adobe). Histograms were created for statistically significant mean metal intensity (MMI) expressions using FlowJo. All representative histograms in the manuscript are chosen from patients with the closest values to the mean of their group. Heatmaps showing z-score normalized median expression levels were created using R package pheatmap, all correlations were performed in R using the packages ggcorrplot, principal component analyses were performed using MMIs for each sample using the stats package in R.

## Statistical analysis and reproducibility

All statistical comparisons were made in GraphPad Prism 10.2.1 (GraphPad Software, San Diego, California USA, www.graphpad.com). For all animal experiments, numbers of biologically independent mice were provided. For all in vitro experiments, numbers of biologically independent samples used in each experiment are provided. Comparison between two groups was examined by two-tailed Student's $t$-tests or two-tailed Mann–Whitney $U$-tests. The log-rank test was used to assess significant differences between survival curves. No statistical method was used to predetermine sample size. No data were excluded from the analyses. Randomization was used for all animal experiments. The Investigators were not blinded to allocation during experiments and outcome assessment, besides measurements of max cristae width which were performed in a blinded fashion to prevent biased outcomes. For all cases, statistical significance was set as $p < 0.05$. All statistical tests used in this study are described in detail in the corresponding figure legends. Results shown represent mean or mean ± standard error of the mean (SEM), as indicated. *$p < 0.05$; **$p < 0.01$; ***$p < 0.001$; ****$p < 0.0001$; ns not significant.

## Reporting summary

Further information on research design is available in the Nature Portfolio Reporting Summary linked to this article.

# Data availability

The RNA sequencing data generated in this study are available at the Gene Expression Omnibus (GEO) repository of the National Center for Biotechnology Information under accession code GSE298663. Metabolomic profiles are available at the NIH Common Fund's National Metabolomics Data Repository (NMDR) website, the Metabolomics Workbench[42] (https://www.metabolomicsworkbench.org, Study IDs: ST003963, ST003964, ST003966. Supplementary information, including Supplementary Figs. 1–8 and Supplementary Data files 1–6 are provided with the online version of this paper. All other datasets generated during and/or analyzed during this study are available from the corresponding author on request. Source data are provided with this paper.

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

## Acknowledgements

Research reported in this publication included work performed in the Integrative Genomics Core and the Integrated Mass Spectrometry Shared Resource supported by the National Cancer Institute of the National Institutes of Health under grant number P30CA033572. The content is solely the responsibility of the authors and does not necessarily represent the official views of the National Institutes of Health. L.G. acknowledges support from NIH K12 grant no. 5K12CA001727–29, the Hyundai Hope on Wheels Young Investigator Award, the Margaret E. Early Medical Research Trust Award, the Schwartz Accelerator Fund, the Norman and Sadie Lee Foundation, and the Albert and Bettie Sacchi Foundation. The authors thank Chunyan Zhang for her meticulous technical support throughout the preparation of this manuscript.

## Author contributions

L.G., S.J.F., and X. Wang. conceived and conceptualized the study. L.G. performed the in vitro and in vivo experiments, analyzed the data, and wrote the manuscript. E.R.H. performed cytometry data analysis and

generated plots. J.W., B.G., R.U., V.V., and R.E. performed in vitro and in vivo experiments and analyzed data. K.V.P., N.P.H., and P.P. performed and analyzed the mass spectrometry experiments. J.S. and J.L.F. performed the extracellular flux experiments. R.Z.N. and Z.L. performed transmission electron microscope experiments. D.W., E.T., and R.K.G. analyzed transmission electron microscope and mass spectrometry data. M.H.C. and X. Wu. analyzed RNA data. T.M., S.B., J.R.W., J.P., M.C.C., D.N., and I.A. collected and analyzed patient data. L.G. and M.C.C. wrote the original draft. All authors reviewed and edited the final manuscript. L.G., X. Wang., and S.J.F. provided resources, acquired funding, and supervised the study.

## Competing interests

The authors declare no competing interests.
