## [Transparent Peer Review file · Nature Communications]

Immunometabolic determinants of long-term response in leukemia patients receiving CD19 CAR T cell therapy

Corresponding Author: Dr Lior Goldberg

Version 0:

Reviewer comments:

Reviewer #1

(Remarks to the Author)

Goldberg et al studied the immunometabolic profile of CAR.CD19 T cells from STR and LTR patients to elucidate the mechanism underlying the success and durability of anti-tumour therapy in these two cohorts of patients. Interestingly, they showed that the difference between CAR T cell DPs derived from STR and LTR patients is not reflected in any surface phenotypic markers or differentiation state, but in the metabolic patterns of the cells. Indeed, they demonstrated that CAR T from LTR patients show increased OXPHOS and increased PPP activity associated with low expression of mTOR.

Consistently, inhibition of mTOR by rapamycin induced a metabolic reprogramming in CAR-Ts that enhanced the anti-tumour activity of CAR-T cells in an in vivo xenograft mouse model, suggesting a therapeutic strategy to improve the long-term success of CAR.CD19 T cell therapy in leukemia.

The work is very interesting for the field, and with a large translational applicability. Nevertheless, data presentation is lacking of details that is limiting the overall value and clarity of the paper itself.

MAIN points

1. The immunometabolic profile of the CAR-negative T cell fraction is not provided in the paper, despite the fact that the authors could likely access this data given the gating strategy they used and described. It would be important to determine whether the differences observed in the CAR+ T cell DP fraction of STR and LTR patients are also evident in CAR- cells, or whether the distinct immunometabolic profile is specific to the T cells expressing the CAR construct.
2. The authors should present and discuss any differences in the immunometabolic profile of CAR T cells in the day 28 bone marrow of STR vs. LTR patients (e.g., through a volcano plot). It would be useful to determine whether the differential profile observed at the DP level is still apparent following DP infusion in the two patient cohorts.
3. The authors should add plots in Figures 1b–e showing ECAR in response to mitochondrial inhibitors in order to evaluate the contribution of glycolysis under mitochondrial stress conditions. Additionally, values of spare respiratory capacity (SRC) should be included to provide a more comprehensive metabolic assessment.
4. Figure 1K only shows data for CD8 cells. The authors should also present the expression patterns observed in CD4 cells for a more complete analysis.
5. The exemplificative CyTOF histogram plots in Figure 1K show very minor differences between STR and LTR samples for several markers (e.g., CPT1A, LAL, and mTOR), despite the single data plots indicating statistically significant differences. A similar discrepancy is observed in Figure 4n. The authors should address these apparent inconsistencies.
6. Given the increased mitochondrial metabolism and pentose phosphate pathway (PPP) activity demonstrated in CAR T cells from LTR patients, it would be highly interesting to assess oxidative stress by measuring ROS levels. This could further support the observed increase in GSH levels, a key antioxidant defense.
7. The MitoTracker data presented in Figure 2 are intriguing, but the observation of a double peak is unconventional. The authors should investigate whether this double peak is also present in the CAR-negative fraction and in primary unstimulated T cells. Furthermore, it would be useful to clarify whether the low MitoTracker fraction differs from the unstained fraction, which is not shown in the plots of Extended Figure 1. Adding measures of mitochondrial membrane polarization to Figures 2 and 4 would also provide a clearer understanding of the mitochondrial capacity of the cells under investigation.
8. There is an inconsistency between the data in Figure 4 and Figures 1 and 2. Specifically, the DP manufactured from STR patient apheresis shows a high MitoTracker peak in Figure 4m, while none of the STR patients in Figure 2b exhibit this feature. Additionally, the immunometabolic profile in Figures 4e–f differs substantially from that shown in Figures 1c and 1e.

These discrepancies should be addressed and clarified.

9. The use of rapamycin to block the mTOR pathway during CAR T cell manufacturing is compelling, but the lack of a significant reduction in T cell proliferation is unexpected, given the well-established role of rapamycin in inhibiting T cell cycle progression and inducing T cell anergy. The authors should provide cell cycle analysis of DP manufactured with and without rapamycin to support their data. Moreover, they should explain the increased anti-tumor activity of DP manufactured with rapamycin despite the apparent metabolic rest, characterized by decreased mitochondrial and glucose metabolism. This finding seems to contradict earlier results suggesting that increased PPP flux and oxidative metabolism in LTR patient DPs are associated with better outcomes.

10. In Figure 5, there appears to be no difference between the cohorts of mice receiving DP from STR and LTR patients (rapamycin-negative). The authors should present the immunometabolic profile of the DPs used in the animal models to determine whether the DP from STR patients differs from that of LTR patients.

11. The study focuses on a specific DP manufactured from PBMCs depleted of CD14+ monocytes and CD25+ Tregs, followed by positive selection of CD62L+ T cells. This non-standard approach should be clearly highlighted to ensure readers understand that the findings are specific to this particular product and should not be generalized without further experimental support.

Minor points

Figures improvements:

1. In general, figure legend captions should be standardized based on the journal requirement (introductory summary sentence in each figure legend, way of listing the figure panels, etc).
2. In general, legends of each figure are not providing enough details to the readers.
3. Figures for cytofluorimetric/Cytof analysis do not report color legend, in both figure and legend.
4. The text does not report the acronym of MMI to what refers to.
5. The DPs are represented by a mix of CAR+ and CAR- T cells. The figures and text should report when the data are presented on a total CAR T cell DP or on the CAR+ T cell fraction isolated by a specific flow-cytometry/Cytof gating.
6. Line 342: it is not clear the reference to non-CAR T cells in leukapheresis samples. Data from leukapheresis samples are not provided in the paper, although they could be very interesting to be included.
7. Authors should justify why they have used two different statistical tools to analyze data in Figure 2b and 2e. Considering the mean and sd of data in figure 2e, the difference appears not of main relevance.

Text improvements:

Line 99: Since figure 1i shows representative histograms and analysis of statistically significant proteins in CD3+ CAR T cells, the text needs to be aligned since it refers to CD4 and CD8.

Line 102-104: Authors should move this part referring to the fatty acids oxidation to line 119 to maintain conceptual continuity with regard to FAO.

line 199: In Figure 4C a % is given, whereas the text refers to "CD4/CD8 ratio" in line 199.

line 199: probably the authors meant mTOR and not rapamycin in the sentence " In line with the role of rapamycin,".

line 319-329: please, include the kit used for Seahorse analysis.

line 377: misspelling for MACSQuant, not MASCQuant

Supplementary Table S2: The authors should include the reference of this table in the Materials and methods section, Cytof paragraph.

Reviewer #2

(Remarks to the Author)

Reviewer #3

(Remarks to the Author)

In this manuscript, Goldberg et al. systematically compared the immunometabolism of CD19-CAR T cells from long-term (LTR) and short-term responders (STR), analyzing their characteristics pre- and post-infusion. To elucidate the immunometabolic features underlying their differential therapeutic efficacy, they applied a comprehensive set of approaches, including functional metabolic assays, mitochondrial morphological assessments, and bulk and single-cell proteomic analyses. Based on these findings, the researchers explored a potential strategy to enhance CAR T cell function by transiently inhibiting mTOR activity with rapamycin during manufacturing. This approach was tested in murine tumor models, showing the ability to improve anti-tumor efficacy.

Overall, the study is well-designed, employing a multi-layered approach to characterize the distinct features of CAR-T cells from long-term and short-term responders. The reported results are comprehensive. However, the strategy used to reprogram the metabolism of CAR-T cells lacks novelty. mTOR and rapamycin have been extensively studied in the past for improving CAR-T cell performance (e.g. PMID:39932788; PMID:34233960; PMID:30890531; PMID: 38014236), with both preclinical mouse models and clinical trials demonstrating their potential to improve anti-tumor efficacy.

There are several issues to address before publication.

1. In Fig. 1h, CD27 is identified as a memory-associated gene, which is expressed at a higher level in STR group. This seems contradictory to the current understanding. Does the memory-associated signature play a positive or negative role in long-term response for CAR-T cell therapy?
2. In Extended Data Fig. 2a-b, the authors measured the different T cell subsets and found no significant differences of the pre-infusion products. How about the T cell subsets in BM?
3. In Fig. 1h, TIGIT and TIM-3, both exhaustion-associated genes, are highly expressed in the LTR group. Similarly, as shown in the post-infusion data (Fig 3c), TIM-3 expression is higher in the CAR-T product. However, in BM, there is no significant difference in PD-1 or TIM-3 expression. Could the BM environment influence and potentially reverse the exhaustion state of CAR-T cells?
4. In Fig. 3c, in the STR group, the CAR-T cells in product have higher TIM-3 expression, while the CAR-T cells in BM have higher PD-1 expression instead of TIM-3. Therefore, it's not obvious to claim that STR acquired an exhaustion phenotype.
5. In Fig. 4i, MYC was significantly upregulated, yet MYC targets were significantly downregulated after rapamycin treatment. There is research showing that rapamycin can also target STAT3 independent of mTOR to suppress tumor growth (PMID:34706270). Therefore, the improved outcome of the anti-tumor effects may not be solely due to the mTOR pathway. Did the authors measure the STAT3 activity?
6. In Fig. 5d and 5i, why did CAR-T cells from STR and LTR exhibit no difference in anti-tumor efficacy in the absence of Rapamycin? Moreover, upon Rapamycin treatment, why do CAR-T cells from LTR show even reduced efficacy as compared to STR+ Rapamycin? Does this tumor model fail to reflect the patient response?

Version 1:

Reviewer comments:

Reviewer #1

(Remarks to the Author)

The Authors had carefully revised the manuscript with the inclusion of the majority of the suggested experiments. The overall clarity and robustness of the manuscript has been significantly improved.

One main issue remains the first animal model (NALM6) that seems do not mirror the differential anti-leukemia activity of LTR and STR as observed in patients, and thus casting doubt on the suitability of the model for the message that the Authors want to give in this paper. Moreover, for this model it is still not clear the reason why the addition of the rapamycin to the manufacturing of the LTR product is significantly worsening the antileukemia activity. Indeed, in the Nalm6 mice experiment, the Authors have pointed out the role of a lower expression of CD69 in STR with rapamycin compared to LTR with rapamycin, suggesting that the less activated CAR T cells enhance anti-tumor activity. It is very difficult to understand why this is happening (in this findings rapamycin is associated to a reduction of CD69 only in STR and not in LTR), and why only in NALM6 model and not in O18z model.

Lastly, considering the revised version of the paper, we suggest to refine the discussion section:

- 1) Pointing out that the rapamycin exposure during the manufacturing is transient and limited to day4-6 (line 340).
- 2) For the CAR T cells exposed to rapamycin, the improved expansion is observed only for the NALM6 model and STR cells (line 335).

Reviewer #2

(Remarks to the Author)

Reviewer #4

(Remarks to the Author)

The authors have articulated the novelty of the manuscript more clearly and have addressed most of the previous comments. The overall quality of the manuscript has been improved. There are a few minor issues that still require clarification before publication.

1. The authors have clarified the CD27 functions not only as a memory-associated maker but also as an effector-related maker. It's still unclear that if the memory-associated genes play a role in CAR T products from LTR and STR (related to previous comments #1)
2. Please label the genes that showed statistical significance in Fig. 1j similar to Fig. 4n, to enhance the readability of the figure.

Version 2:

Reviewer comments:

Reviewer #1

(Remarks to the Author)

We thank the authors for their final revision. The reviewer has no further comments.

Reviewer #2

(Remarks to the Author)

Reviewer #4

(Remarks to the Author)

I thank the authors for their responses to my previous comments. The revisions have adequately addressed my concerns, and the manuscript has been substantially improved as a result. Overall, I find the responses satisfactory and believe that the current version of the manuscript meets the standards for publication.

Manuscript number: NCOMMS-25-11102-T

Title: Immunometabolic determinants of long-term response in leukemia patients receiving CD19 CAR T cell therapy

Reviewer Comments:

Reviewer #1

Goldberg et al studied the Immunometabolic profile of CAR.CD19 T cells from STR and LTR patients to elucidate the mechanism underlying the success and durability of anti-tumour therapy in these two cohorts of patients. Interestingly, they showed that the difference between CAR T cell DPs derived from STR and LTR patients is not reflected in any surface phenotypic markers or differentiation state, but in the metabolic patterns of the cells. Indeed, they demonstrated that CAR T from LTR patients show increased OXPHOS and increased PPP activity associated with low expression of mTOR. Consistently, inhibition of mTOR by rapamycin induced a metabolic reprogramming in CAR-Ts that enhanced the anti-tumour activity of CAR-T cells in an in vivo xenograft mouse model, suggesting a therapeutic strategy to improve the long-term success of CAR.CD19 T cell therapy in leukemia.

The work is very interesting for the field, and with a large translational applicability. Nevertheless, data presentation is lacking of details that is limiting the overall value and clarity of the paper itself..

We thank the Reviewer for the positive comments on our work and the careful evaluation of our data and analyses.

MAIN points

1. The immunometabolic profile of the CAR-negative T cell fraction is not provided in the paper, despite the fact that the authors could likely access this data given the gating strategy they used and described. It would be important to determine whether the differences observed in the CAR+ T cell DP fraction of STR and LTR patients are also evident in CAR- cells, or whether the distinct immunometabolic profile is specific to the T cells expressing the CAR construct.

Response: We agree that comparing the CAR-negative fraction of STR and LTR patients may also be informative. As suggested by the Reviewer, we now include a detailed CyTOF analysis of the CAR-negative T cell fraction in the revised text (paragraph 1, page 4) and in Supplementary Fig. 2a-b.

“Lastly, we leveraged our single-cell mass cytometry approach (Supplementary Fig. 1) and compared the CAR-negative T cells in the pre-infusion products of both groups (Supplementary Fig. 2a). Similar to CD3 CAR-positive T cells, CD3 CAR-negative T cells of LTR exhibited significantly increased levels of HK, G6PD, and CPT1A compared to STR (Supplementary Fig. 2b), suggesting these immunometabolic changes are independent of CAR expression.”

Supplementary Fig. 2: CD3⁺ CAR⁻ T cells of long-term responders show increased mitochondrial mass and expression of key enzymes involved in oxidative phosphorylation, fatty acid oxidation, and pentose phosphate pathway activity. **a** Heat map indicating z-score normalized median expression of all CyTOF markers assessed in CD4⁺ and CD8⁺ CAR⁻ T cells. **b** Representative histograms and analysis of statistically significant proteins in CD3⁺ CAR⁻ T cells.

2. The authors should present and discuss any differences in the immunometabolic profile of CAR T cells in the day 28 bone marrow of STR vs. LTR patients (e.g., through a volcano plot). It would be useful to determine whether the differential profile observed at the DP level is still apparent following DP infusion in the two patient cohorts.

Response: As proposed by the Reviewer, we present and discuss any differences in the immunometabolic profile of CAR T cells in day 28 bone marrow of STR vs. LTR patients in the revised text (page 6) and in Fig. 3c.

“Similar to the pre-infusion products, CD3⁺ CAR⁻ T cells in the BM at day 28 post-infusion of STR showed significantly higher expression of CD98 and ATP5a, which is consistent with effector activation¹². However, CD3⁺ CAR⁻ T cells of LTR exhibited significantly increased mTOR and pS6 expression compared to STR, suggesting a shift from a memory-to-effector metabolic phenotype of LTR CAR⁻ T cells from pre- to post-infusion, respectively (Fig. 3c). Indeed, longitudinal comparison of STR and LTR CAR⁻ T cells pre-infusion vs. post-infusion revealed that pre-infusion...”

Fig. 3: CAR T cells in post-treatment bone marrow of long-term responders show increased mTOR signaling. **c** Metabolic state PCA of CD3⁺ CAR⁺ T cell products and CD3⁺ CAR⁺ T cells in the day 28 bone marrow, considering only the expression of metabolic proteins.

c Representative histograms and analysis of statistically significant proteins in CD3+ CAR+ T cells in day 28 bone marrow.

3. The authors should add plots in Figures 1b–e showing ECAR in response to mitochondrial inhibitors in order to evaluate the contribution of glycolysis under mitochondrial stress conditions. Additionally, values of spare respiratory capacity (SRC) should be included to provide a more comprehensive metabolic assessment.

Response: As suggested by the Reviewer, we added the ECAR and SRC data to Figure 1.

Furthermore, based on the Reviewer's suggestion, we have added the ECAR data to Figure 4.

4. Figure 1K only shows data for CD8 cells. The authors should also present the expression patterns observed in CD4 cells for a more complete analysis.

Response: We agree that the expression patterns observed in CD4 cells would present a more complete analysis. However, due to a lack of page space, we show in Figure 1m (former Figure 1K) the pertinent and statistically significant results. All of the other data, including CD4+ CAR+, can be found in the Source Data file. We have included for the Reviewer an analysis of the CD4+ CAR+ cells, which shows a similar biological trend to that observed in the statistically significant proteins of CD8+ CAR+ cells.

Reviewers only Figure 1. Analysis of CD89, LAL, and mTOR expression in CD4+ CAR+ T cells.

5. The exemplificative CyTOF histogram plots in Figure 1K show very minor differences between STR and LTR samples for several markers (e.g., CPT1A, LAL, and mTOR), despite the single

data plots indicating statistically significant differences. A similar discrepancy is observed in Figure 4n. The authors should address these apparent inconsistencies.

Response: We appreciate the Reviewer's observation regarding the representative histogram and understand that some of the representative histogram plots may appear to show only very minor differences. As shown in the full analysis, with n=8 in each group, the markers in Figure 1m (formerly Figure 1K) and Figure 4r (formerly Figure 4n) are statistically significantly different. In the corresponding representative plots, we deliberately selected histograms from patients with the closest value to the mean of their respective groups (i.e., truly representative). We respectfully disagree that these examples represent discrepancies in the data. We have clarified this point in the revised text, Methods/Visualization (page 16).

"All representative histograms in the manuscript are chosen from patients with the closest values to the mean of their group."

We have included for the Reviewer an example of our representative histogram, where the individual histograms were chosen from patients with the closest value to the mean, compared to a selective histogram, where the individual histograms were chosen from the highest and lowest expressing patients, of CPT1A. Both of these data points were included in the corresponding data plots with n=8 for each group and contributed to the statistical significance we reported.

Reviewers only Figure 2. Histograms of CPT1A expression. Left-representative histograms showing patients closest to the mathematical mean of their group (STR/LTR). Right-selective histograms choosing from patients representing the highest and lowest expressing of their group (STR/LTR).

6. Given the increased mitochondrial metabolism and pentose phosphate pathway (PPP) activity demonstrated in CAR T cells from LTR patients, it would be highly interesting to assess oxidative stress by measuring ROS levels. This could further support the observed increase in GSH levels, a key antioxidant defense.

Response: As proposed by the Reviewer, we conducted new experiments and assessed oxidative stress by measuring reactive oxygen species levels and added it to the revised text (page 5) and in Supplementary Fig. 3.

"Lastly, given the increased PPP activity observed in CAR T cells of LTR and the role of this metabolic pathway in maintaining cellular redox balance, we measured the reactive oxygen species levels. We found no difference between the two groups (Supplementary Fig. 3), supporting our previous metabolic assays showing that PPP activity mainly supports fatty acid oxidation, the TCA cycle, and nucleotide synthesis."

Supplementary Fig. 3: Reactive oxygen species assessment by flow cytometry. a Representative of gating strategy. All samples were FSC and SSC gated. DAPI negative cells were then gated, followed by FSC-A/FSC-H gating to select singlet cells. CAR T cells were detected using CD3 and EGFR staining, and reactive oxygen species level was assessed by measuring the CellROX fluorescent probe expression (%). **b** Analysis of CellROX expression in CD4+ and CD8+ CAR+ T cells products.

7. The MitoTracker data presented in Figure 2 are intriguing, but the observation of a double peak is unconventional. The authors should investigate whether this double peak is also present in the CAR-negative fraction and in primary unstimulated T cells. Furthermore, it would be useful to clarify whether the low MitoTracker fraction differs from the unstained fraction, which is not shown in the plots of Extended Figure 1. Adding measures of mitochondrial membrane polarization to Figures 2 and 4 would also provide a clearer understanding of the mitochondrial capacity of the cells under investigation.

Response: We appreciate the Reviewer's observation regarding the bimodal distribution of MitoTracker staining in patients' CAR T cells. This bimodal pattern is a well-recognized phenomenon and reflects metabolic heterogeneity within the T cell populations. Similar bimodal patterns have been reported in both murine and human T cells, particularly during or after activation or in different tissues. (Sukumar et al., *Nature*, 2013; Buck et al., *Immunity*, 2016; Villa, et al, *Nat Commun.* 2024; Beckermann et al, *JCI Insight.* 2020). We have included four supporting references and two representative figures to illustrate further and support this observation.

Reviewers only Figure 3. Bi-model distribution of T cells stained with Mitotracker green; adapted from Figure 2b in Villa M, Sanin DE, Apostolova P, Corrado M, Kabat AM, Cristinzio C, Regina A, Carrizo GE, Rana N, Stanczak MA, Baixauli F, Grzes KM, Cupovic J, Solagna F, Hackl A, Globig AM, Hässler F, Puleston DJ, Kelly B, Cabezas-Wallscheid N, Hasselblatt P, Bengsch B, Zeiser R, Sagar, Buescher JM, Pearce EJ, Pearce EL. Prostaglandin E2 controls the metabolic adaptation of T cells to the intestinal microenvironment. Nat Commun. 2024 Jan 11;15(1):451. doi: 10.1038/s41467-024-44689-2. PMID: 38200005.

Reviewers only Figure 4. Bi-model distribution of T cells stained with Mitotracker green; adapted from Figure 6b in Beckermann KE, Hongo R, Ye X, Young K, Carbonell K, Healey DCC, Siska PJ, Barone S, Roe CE, Smith CC, Vincent BG, Mason FM, Irish JM, Rathmell WK, Rathmell JC. CD28 costimulation drives tumor-infiltrating T cell glycolysis to promote inflammation. JCI Insight. 2020 Aug 20;5(16):e138729. doi: 10.1172/jci.insight.138729. PMID: 32814710.

As suggested by the Reviewer, we investigated whether this bi-model distribution is present in the CAR-negative fraction. We added it to the revised text (page 5) and in Supplementary Fig. 2c.

“A similar finding was observed in CD3 CAR T-negative cells, suggesting these mitochondrial changes are independent of CAR expression (Supplementary Fig. 2c).”

Supplementary Fig. 2: CD3⁺ CAR⁻ T cells of long-term responders show increased mitochondrial mass and expression of key enzymes involved in oxidative phosphorylation,

fatty acid oxidation, and pentose phosphate pathway activity. c Analysis of MitoTracker expression in CD4+ and CD8+ CAR- T cells products.

We apologize for the lack of clarity. Supplementary Figure 4 shows the suggested unstained/fluorescence minus one (highlighted in red).

As proposed by the Reviewer, we conducted new experiments and measured mitochondrial membrane polarization in the patients' pre-infusion CAR T cells (Figure 2) and discuss it in the revised text (page 5) and Supplementary Fig. 6.

“Lastly, to further evaluate the mitochondrial features of the CAR T cells, we measured the mitochondrial membrane potential and found no differences between the two groups (Supplementary Fig. 6).”

Supplementary Fig. 6: Mitochondrial membrane potential assessment by flow cytometry. **a** Representative of gating strategy. All samples were FSC and SSC gated, followed by FSC-A/FSC-H gating to select singlet cells. DAPI negative cells were then gated, and CAR cells were detected using CD3 and EGFR staining. Mitochondrial membrane potential was assessed by measuring the TMRM (Tetramethylrhodamine methyl ester) dye expression (%). **b** Analysis of TMRM expression in CD4+ and CD8+ CAR+ T cells pre-infusion products.

Unfortunately, due to the scarcity of patient clinical samples, it was not feasible to assess mitochondrial membrane polarization in the CAR T cells from bone marrow samples.

8. There is an inconsistency between the data in Figure 4 and Figures 1 and 2. Specifically, the DP manufactured from STR patient apheresis shows a high MitoTracker peak in Figure 4m, while none of the STR patients in Figure 2b exhibit this feature. Additionally, the immunometabolic profile in Figures 4e–f differs substantially from that shown in Figures 1c and 1e. These discrepancies should be addressed and clarified.

Response: We appreciate the Reviewer’s observation regarding the immunometabolic differences observed between the data presented in Figures 1 and 2 (pre-infusion CAR T cells) and Figure 4 (patient-derived CAR T cells with and without rapamycin). By focusing in only on modulating mTOR activity with rapamycin, we did not expect to recapitulate in STR-derived products the immunometabolic profile of LTR pre-infusion CAR T cell products and, therefore, respectfully disagree that this represents a discrepancy. We have clarified that our goal was to isolate the effects of mTOR activity in STR-derived products on page 7 of the manuscript.

“Intrigued by our pre- and post-infusion results suggesting a significant role for mTOR activity on the efficacy of CAR T cells of LTR, we asked whether we could harness the mTOR pathway to enhance anti-tumor activity of CAR T cells of STR. To that end, we manufactured CAR T cells from STR-derived leukapheresis products with or without the mTOR inhibitor rapamycin to transiently inhibit mTOR activity with the goal of isolating the effects of mTOR signaling in the STR-derived products.”

As highlighted in the main text, the observation of lower mTOR expression in LTR compared to STR in the pre-infusion CAR T cells suggested that lower mTOR expression is related to improved outcome. mTOR immunometabolic reprogramming is a continuum (as highlighted in **Reviewers only Figure 5**). Therefore, we do not argue that by adding rapamycin to STR we would fully recapitulate the immunometabolic profile of LTR. Indeed, the mTOR MMI percent reduction in CAR T cell products of LTR compared to STR is -15%, while the mTOR MMI percent reduction in STR with rapamycin compared to STR without rapamycin CAR T cells is -34%. This would explain the differences in the immunometabolic profile between Figures 4e-f and Figures 1c,e.

REDACTED

Reviewers only Figure 5. Adapted from Figure 1 in Shi H, Chen S, Chi H. Immunometabolism of CD8+ T cell differentiation in cancer. Trends Cancer. 2024 Jul;10(7):610-626. doi: 10.1016/j.trecan.2024.03.010. Epub 2024 Apr 30. PMID: 38693002.

The observed differences in the high MitoTracker peak levels between Figure 2b and Figure 4m are likely related to differences in the manufacturing scale between clinical-grade manufacturing and lab-based manufacturing. Although both follow the same timeline and use the same reagents, clinical products are made in large bags, producing hundreds of millions of CAR T cells, while lab-based products are made in plates/flasks to produce a few million CAR T cells. Nevertheless, the biological result is similar; lower mTOR expression results in an increase in the MitoTracker high fraction. Furthermore, Supplementary Figure 6 shows that STR CAR T cell products have a high peak fraction up to 9.4% and is not absent.

9. The use of rapamycin to block the mTOR pathway during CAR T cell manufacturing is compelling, but the lack of a significant reduction in T cell proliferation is unexpected, given the well-established role of rapamycin in inhibiting T cell cycle progression and inducing T cell anergy. The authors should provide cell cycle analysis of DP manufactured with and without rapamycin to support their data. Moreover, they should explain the increased anti-tumor activity of DP manufactured with rapamycin despite the apparent metabolic rest, characterized by decreased mitochondrial and glucose metabolism. This finding seems to contradict earlier results suggesting that increased PPP flux and oxidative metabolism in LTR patient DPs are associated with better outcomes.

Response: As suggested by the Reviewer, we conducted new experiments and investigated the cell-cycle distributions in CAR T cells manufactured with rapamycin. We added this analysis to the revised text (pages 7-8) and in Fig. 4d-e.

“Intrigued by the similarity in the expansion of both groups and given the immunosuppressive function of rapamycin by inhibiting entry into the cell cycle, we measured the cell cycle distributions of both groups. We observed that on day 11, most CAR T cells in both groups exited the non-dividing Go stage and found slight significant differences between the two groups in the

S and G0/G1 phases (Fig. 4d-e), suggesting that starting to add rapamycin on day 4 post-anti-CD3/CD28 beads activation did not prevent activated CAR T cells from dividing.”

Fig. 4: Transient mTOR inhibition induces metabolic and activation rest and improves metabolic fitness of CAR T cell products derived from short-term responders. d Representative biaxial plots to measure cell cycle phases in CD3+ CAR+ T cells. **e** Percentage of CD3+ CAR+ T cells in each cell cycle phase as measured by mass cytometry.

Regarding the metabolic rest, as discussed above (reviewer 1, comment 8), we do not argue that by adding rapamycin to STR, we fully recapitulate every aspect of the immunometabolic profile of LTR, given the differences in the level of inhibition.

10. In Figure 5, there appears to be no difference between the cohorts of mice receiving DP from STR and LTR patients (rapamycin-negative). The authors should present the immunometabolic profile of the DPs used in the animal models to determine whether the DP from STR patients differs from that of LTR patients.

Response: We appreciate the Reviewer’s observation regarding the differences in survival of mice receiving CAR T cell products derived from STR and LTR patients (rapamycin-negative). However, we cannot directly compare the results from mice treated with different patient products due to known product variability. To our knowledge, there is no well-established mouse model that allows precise comparison and prediction of response between patients’ CD19 CAR T cell products, given the complexity and multifactorial nature of humans as well as the limitations of different animal models (discussed in Duncan BB, Dunbar CE, Ishii K. Applying a clinical lens to animal models of CAR-T cell therapies. *Mol Ther Methods Clin Dev.* 2022 Aug 30;27:17-31. doi: 10.1016/j.omtm.2022.08.008. PMID: 36156878). Therefore, although tempting, it is misleading to make such a comparison.

To determine whether the lack of difference in survival between the cohorts of mice receiving CAR T cells from STR and LTR patients was due to the leukemia model or whether the lack of difference was due to the CAR T cells themselves, we decided our hypothesis in a second leukemia model (central nervous system [CNS] leukemia model using 018z patient-derived cell line; Figure 6). In the Nalm6 leukemia model we included in the initial manuscript, mice treated with STR without rapamycin CAR T cells had slightly longer non-significant survival compared to LTR without rapamycin (Fig. 5d and i) and in the 018z model, mice treated with LTR without rapamycin CAR T cells had slightly longer non-significant survival compared to STR without rapamycin (Fig. 6d, h). Therefore, while the models used may slightly affect the results, they do not alter the conclusions.

REDACTED

Reviewers only Figure 6. Summary of the survival plots in the two leukemia models presented (Nalm6 and 018z).

We have revised the description of the mouse models in the manuscript text on page 9 and added the CNS leukemia model as Figure 6.

“Similar results were observed in a central nervous system (CNS) B-ALL model, where NSG mice were engrafted with 018z leukemic cells, derived from a child with suspected CNS involvement leukemia¹⁹ (Fig. 6a, d). We observed a significant reduction in tumor burden and prolonged survival in mice injected with rapamycin-treated CAR T cell products compared with untreated CAR T cells from both STR (Fig. 6b-c) and LTR (Fig. 6e-f).”

11. The study focuses on a specific DP manufactured from PBMCs depleted of CD14+ monocytes and CD25+ Tregs, followed by positive selection of CD62L+ T cells. This non-standard approach should be clearly highlighted to ensure readers understand that the findings are specific to this particular product and should not be generalized without further experimental support.

Response: The manufacturing methods, while non-standard in the broader CAR T cell field, are used in several of our current clinical manufacturing platforms at City of Hope, including our CD19 CAR T cell manufacturing platform. As proposed by the Reviewer, we have included the manufacturing method as a potential limitation of our study in the revised discussion (page 11).

“Finally, our findings are based on a manufacturing approach that depletes leukapheresis products of CD14+ monocytes and CD25+ Tregs, followed by positive selection of CD62L+ T cells, and should not be generalized to other manufacturing approaches without further experimental support.”

Minor points

Figures improvements:

1. In general, figure legend captions should be standardized based on the journal requirement (introductory summary sentence in each figure legend, way of listing the figure panels, etc).

Response: We appreciate the recommendation and have revised the manuscript according to the journal's requirements.

2. In general, legends of each figure are not providing enough details to the readers.

Response: We appreciate the recommendation and have revised the manuscript to include more detailed information.

3. Figures for cytofluorimetric/Cytof analysis do not report color legend, in both figure and legend.

Response: We appreciate the recommendation and have revised the manuscript.

4. The text does not report the acronym of MMI to what refers to.

Response: We appreciate the recommendation and have defined the acronym in the legend of Figure 1 and Figure 3 as well as in the Methods/Visualization (page 16).

5. The DPs are represented by a mix of CAR+ and CAR- T cells. The figures and text should report when the data are presented on a total CAR T cell DP or on the CAR+ T cell fraction isolated by a specific flow-cytometry/Cytof gating.

Response: We appreciate the recommendation and have revised the manuscript throughout to report accordingly.

6. Line 342: it is not clear the reference to non-CAR T cells in leukapheresis samples. Data from leukapheresis samples are not provided in the paper, although they could be very interesting to be included.

Response: We thank the Reviewer for spotting this detail; the text regarding leukapheresis samples was omitted.

7. Authors should justify why they have used two different statistical tools to analyze data in Figure 2b and 2e. Considering the mean and sd of data in figure 2e, the difference appears not of main relevance.

Response: We appreciate the Reviewer's observation regarding the different statistical tools used to analyze Figures 2b and 2e. We used a two-tailed Mann–Whitney U-test and not a two-tailed unpaired Student's t-test in Figure 2e since the data is not normally distributed; therefore, a Student's t-test is not recommended. **Reviewers only Figure 6** shows four different distribution tests performed in Prism, indicating that our data is not normally distributed. A similar statistical approach has been reported (Göbel, Jana et al. *Cell metabolism*. 2020; Mondet, Julie et al. *Experimental hematology*. 2021; Siegmund, Stephanie E et al. *iScience* 2018).

Reviewers only Figure 6. Tests for normal distribution.

Normality and Lognormality Tests		A	B
Tabular results		LTR	STR
1	Test for normal distribution		
2	D'Agostino & Pearson test		
3	K2	42.51	30.91
4	P value	<0.0001	<0.0001
5	Passed normality test (alpha=0.05)?	No	No
6	P value summary	****	****
7			
8	Anderson-Darling test		
9	A2*	3.832	3.232
10	P value	<0.0001	<0.0001
11	Passed normality test (alpha=0.05)?	No	No
12	P value summary	****	****
13			
14	Shapiro-Wilk test		
15	W	0.9401	0.9317
16	P value	<0.0001	<0.0001
17	Passed normality test (alpha=0.05)?	No	No
18	P value summary	****	****
19			
20	Kolmogorov-Smirnov test		
21	KS distance	0.1161	0.1181
22	P value	<0.0001	<0.0001
23	Passed normality test (alpha=0.05)?	No	No
24	P value summary	****	****

Text improvements:

Line 99: Since figure 1i shows representative histograms and analysis of statistically significant proteins in CD3+ CAR T cells, the text needs to be aligned since it refers to CD4 and CD8.

Response: We appreciate the recommendation and re-worded the manuscript for better clarification.

Line 102-104: Authors should move this part referring to the fatty acids oxidation to line 119 to maintain conceptual continuity with regard to FAO.

Response: We appreciate the recommendation and have revised the manuscript.

line 199: In Figure 4C a % is given, whereas the text refers to "CD4/CD8 ratio" in line 199.

Response: We appreciate the recommendation and have revised the manuscript for clarity.

line 199: probably the authors meant mTOR and not rapamycin in the sentence " In line with the role of rapamycin,".

Response: We appreciate the recommendation and have corrected the manuscript accordingly.

line 319-329: please, include the kit used for Seahorse analysis.

Response: We appreciate the recommendation and have included the data in the revised manuscript.

line 377: misspelling for MACSQuant, not MASCQuant

Response: We appreciate the recommendation and have corrected the typo.

Supplementary Table S2: The authors should include the reference of this table in the Materials and methods section, Cytof paragraph.

Response: We appreciate the recommendation and have revised the manuscript accordingly.

Reviewer #2

Response: We thank the Reviewer for the positive comments on our work and the careful evaluation of our data and analyses.

Reviewer #3

In this manuscript, Goldberg et al. systematically compared the immunometabolism of CD19-CAR T cells from long-term (LTR) and short-term responders (STR), analyzing their characteristics pre- and post-infusion. To elucidate the immunometabolic features underlying their differential therapeutic efficacy, they applied a comprehensive set of approaches, including functional metabolic assays, mitochondrial morphological assessments, and bulk and single-cell proteomic analyses. Based on these findings, the researchers explored a potential strategy to enhance CAR T cell function by transiently inhibiting mTOR activity with rapamycin during manufacturing. This approach was tested in murine tumor models, showing the ability to improve anti-tumor efficacy.

Overall, the study is well-designed, employing a multi-layered approach to characterize the distinct features of CAR-T cells from long-term and short-term responders. The reported results are comprehensive. However, the strategy used to reprogram the metabolism of CAR-T cells lacks novelty. mTOR and rapamycin have been extensively studied in the past for improving CAR-T cell performance (e.g. PMID:39932788; PMID:34233960; PMID:30890531; PMID: 38014236), with both preclinical mouse models and clinical trials demonstrating their potential to improve anti-tumor efficacy.

We thank the Reviewer for acknowledging the well-designed and multi-layered approach of our work in determining the distinct immunometabolic features of CAR-T cells from long-term and short-term responders, and for the thoughtful critique. Although the use of rapamycin has been previously reported, these manuscripts lack the comprehensive mechanistic insight provided here and are based on healthy donor T cells, which differ from T cells of heavily treated patients. Notably, these studies lack mechanistic connection between rapamycin and the metabolic signature of CAR T cells in patients. The goal of our study is not to examine how rapamycin affects the CAR, but to use rapamycin to validate the metabolic determinants. We are the first to use patient-derived CAR T cells to study the metabolic determinants and their correlation with outcomes.

There are several issues to address before publication.

1. In Fig. 1h, CD27 is identified as a memory-associated gene, which is expressed at a higher level in STR group. This seems contradictory to the current understanding. Does the memory-associated signature play a positive or negative role in long-term response for CAR-T cell therapy?

Response: We appreciate the Reviewer's observation regarding the higher CD27 expression in STR in Figure 1h heatmap. The heatmap indicates z-score normalized median expression of all CyTOF markers assessed in CD8+ and CD4+ CAR T cells. Although the median expression of

CD27 was higher in STR, it did not reach statistical significance. Furthermore, we (Goldberg L, Haas ER, Vyas V, Urak R, Forman SJ, Wang X. Single-cell analysis by mass cytometry reveals CD19 CAR T cell spatiotemporal plasticity in patients. *Oncoimmunology*. 2022 Feb 18;11(1):2040772. doi: 10.1080/2162402X.2022.2040772. PMID: 35223193) and others (Schiött A, Lindstedt M, Johansson-Lindbom B, Roggen E, Borrebaeck CA. CD27- CD4+ memory T cells define a differentiated memory population at both the functional and transcriptional levels. *Immunology*. 2004 Nov;113(3):363-70. doi: 10.1111/j.1365-2567.2004.01974.x. PMID: 15500623), have shown that CD27 can be expressed in both memory and effector T cell subsets and therefore is not a reliable marker for memory phenotype. Based on the reviewer's comment, we decided to revise the manuscript and exclude our subset analysis of CAR T cell products based on CD27.

Reviewers only Figure 6. Adapted from Figure 1 in Goldberg, Lior et al. *Oncoimmunology*. 2022. CD27 shows bimodal distribution in effector and effector memory T cells. TN = naïve T cells; TSCM = stem cell memory T cells; TCM = central memory T cells; TEFF = effector T cells; TEM = effector memory T cells; TEMRA = effector memory CD45RA+ T cells.

2. In Extended Data Fig. 2a-b, the authors measured the different T cell subsets and found no significant differences of the pre-infusion products. How about the T cell subsets in BM?

Response: As suggested by the Reviewer, we now include information on the CAR T cell subsets in day 28 bone marrow in the revised text (page 9) and in Supplementary Fig. 7c.

“Lastly, similar to pre-infusion products, we did not observe significant differences in the abundance of different T cell subsets in CAR T cells from the day 28 bone marrow (Supplementary Fig. 7d).”

3. In Fig. 1h, TIGIT and TIM-3, both exhaustion-associated genes, are highly expressed in the LTR group. Similarly, as shown in the post-infusion data (Fig 3c), TIM-3 expression is higher in the CAR-T product. However, in BM, there is no significant difference in PD-1 or TIM-3 expression. Could the BM environment influence and potentially reverse the exhaustion state of CAR-T cells?

Response: We appreciate the Reviewer's observation regarding the higher TIGIT and TIM-3 expression in LTR in Figure 1i (former Figure 1h) heatmap. The heatmap indicates z-score normalized median expression of all CyTOF markers assessed in CD8+ and CD4+ CAR T cells.

Although the median expression of TIGIT and TIM-3 was higher in LTR, it did not reach statistical significance. Furthermore, immune checkpoint proteins such as TIGIT, TIM-3, and PD-1 expression can be related to both exhaustion and activation and is context-dependent. Indeed, although LTR pre-infusion CAR T cells had overall higher median expression of TIGIT and TIM-3, these CAR T cells led to long-term remission in patients. Therefore, we did not draw any determinantal conclusions regarding the exhaustion state of the CAR T cells in pre-infusion product or post-infusion bone marrow throughout the manuscript. We agree with the Reviewer that the bone marrow microenvironment can influence and potentially reverse exhaustion and/or over-activation state of CAR T cells, and as proposed by the Reviewer, we have included it in the revised text (page 7).

“Furthermore, we found that LTR had significant increases in serine and inosine, which are required for optimal T cell proliferation, enhanced adoptive immunotherapy, and can reverse CAR T cell exhaustion in preclinical models^{14,15}.”

4. In Fig. 3c, in the STR group, the CAR-T cells in product have higher TIM-3 expression, while the CAR-T cells in BM have higher PD-1 expression instead of TIM-3. Therefore, it's not obvious to claim that STR acquired an exhaustion phenotype.

Response: We appreciate the Reviewer's observation regarding the changes in TIM3 and PD-1 expression in LTR CAR T cells pre- and post-infusion. Immune checkpoint protein expression, such as TIGIT, TIM-3, and PD-1, can be related to both exhaustion and activation and is context-dependent; therefore, we did not draw any determinantal conclusions regarding the exhaustion state of the CAR T cells in the manuscript.

“CAR T cells in the day 28 BM of STR had significantly higher expression of PD-1 compared to the pre-infusion product, suggesting that CAR T cells of STR may have acquired an exhausted or over-activated phenotype post infusion (Fig. 3c).”

5. In Fig. 4i, MYC was significantly upregulated, yet MYC targets were significantly downregulated after rapamycin treatment. There is research showing that rapamycin can also target STAT3 independent of mTOR to suppress tumor growth (PMID:34706270). Therefore, the improved outcome of the anti-tumor effects may not be solely due to the mTOR pathway. Did the authors measure the STAT3 activity?

Response: We agree that it is possible that our results may be impacted by the effect of rapamycin on STAT3 activity. Therefore, we conducted and analyzed the STAT3 levels in patient-derived CAR T cell products with and without rapamycin. Our analysis of RNA, pSTAT3 by flow cytometry, and total STAT3, as well as pSTAT3 by western blot, did not reveal statistically significant differences, suggesting that the biology reported in tumor levels is different from that in CAR T cells. Therefore, we are keeping this analysis as reviewer only figure. We agree that the improved outcome of the anti-tumor effects may not be solely due to the mTOR pathway.

Reviewers only Figure 7. STAT3 levels were analyzed by RNA sequencing (a), flow cytometry (b), and western blot (c and d).

6. In Fig. 5d and 5i, why did CAR-T cells from STR and LTR exhibit no difference in anti-tumor efficacy in the absence of Rapamycin? Moreover, upon Rapamycin treatment, why do CAR-T cells from LTR show even reduced efficacy as compared to STR+ Rapamycin? Does this tumor model fail to reflect the patient response?

Response: We appreciate the Reviewer's observation regarding the differences in survival of mice receiving CAR T cell products derived from STR and LTR patients (rapamycin-negative). To our knowledge, there is no well-established mouse model that allows precise comparison and prediction of response between patients' CD19 CAR T cell products, given the complexity and multifactorial nature of humans as well as the limitations of different animal models (discussed in Duncan BB, Dunbar CE, Ishii K. Applying a clinical lens to animal models of CAR-T cell therapies. *Mol Ther Methods Clin Dev.* 2022 Aug 30;27:17-31. doi: 10.1016/j.omtm.2022.08.008. PMID: 36156878). Therefore, although tempting, it might be misleading to make such a comparison.

To determine whether our results were due to our tumor model, we decided to test our hypothesis using a second leukemia model (central nervous leukemia [CNS] model using 018z patient-derived cell line; Figure 6). Indeed, in the Nalm6 leukemia model, mice treated with STR without rapamycin CAR T cells had slightly longer non-significant survival compared to LTR without rapamycin (Fig. 5d and i). While in the 018z CNS leukemia model, we observed the opposite trend (Fig. 6d and h).

Reviewers only Figure 8. Summary of the survival plots in the two leukemia models presented (Nalm6 and 018z).

Nevertheless, based on the Reviewer's comment and intrigued by the differences seen in survival in our Nalm6 model between the STR with rapamycin and LTR with rapamycin, as well as our observation of decreased CAR T cell activation induced by rapamycin, we analyzed the expression level of the activation marker CD69 in both groups. We found lower expression of CD69 in STR compared to LTR, including in the patient-derived CAR T cells used in the Nalm6 mice experiments (Supplementary Fig. 8), suggesting that less activated CAR T cells had enhanced anti-tumor activity.

“Lastly, after showing enhanced anti-leukemic activity in both STR and LTR patient-derived CAR T cells manufactured with rapamycin, in two independent models of leukemia (Nalm6 and 018z), we were intrigued by the improved overall survival of STR with rapamycin compared to the LTR with rapamycin in the Nalm6 model (Fig. 5d, i). Based on our findings of decreased CAR T cell activation induced by rapamycin and the relationship between precise patient-specific control of activation level and CAR T cell functionality²⁰ we assessed CD69 expression in both groups. We found lower expression of CD69 in STR with rapamycin compared to LTR with rapamycin, including in the patient-derived CAR T cells used in the Nalm6 mice experiments (Supplementary Fig. 8), suggesting that the less activated CAR T cells enhance anti-tumor activity.”

Supplementary Fig. 8: Comparison of CD69 expression in CAR T cell products of short- and long-term responders manufactured with Rapamycin. a Representative biaxial plots to measure CD69 expression in CD3⁺ CAR⁺ T cells. **b** CD69 expression in CD3⁺ CAR⁺ T cells products of short- and long-term responders manufactured with Rapamycin. Highlighted dots represent the CD69 expression level and UPN number in CAR T cell products used for the in vivo leukemia model experiment (Fig. 5).

Manuscript number: NCOMMS-25-11102A

Title: Immunometabolic determinants of long-term response in leukemia patients receiving CD19 CAR T cell therapy

Editorial Decision: Revision

Reviewer Comments:

Reviewer #1

The Authors had carefully revised the manuscript with the inclusion of the majority of the suggested experiments. The overall clarity and robustness of the manuscript has been significantly improved.

One main issue remains the first animal model (NALM6) that seems do not mirror the differential anti-leukemia activity of LTR and STR as observed in patients, and thus casting doubt on the suitability of the model for the message that the Authors want to give in this paper. Moreover, for this model it is still not clear the reason why the addition of the rapamycin to the manufacturing of the LTR product is significantly worsening the antileukemia activity. Indeed, in the Nalm6 mice experiment, the Authors have pointed out the role of a lower expression of CD69 in STR with rapamycin compared to LTR with rapamycin, suggesting that the less activated CAR T cells enhance anti-tumor activity. It is very difficult to understand why this is happening (in this findings rapamycin is associated to a reduction of CD69 only in STR and not in LTR), and why only in NALM6 model and not in 018z model.

Response: We agree that the NALM6 model does not mirror the differential anti-leukemia activity of LTR and STR as observed in patients. Recapitulating this clinical observation in our mouse model is not possible given the variability in CAR T cell products from different donors (i.e., patients) and the limitations of mouse models. Therefore, the comparator in this experiment is not STR vs. LTR (as in our observations in patients), but rather STR without rapamycin vs. STR with rapamycin (Fig 5a-e) as well as LTR without rapamycin vs. LTR with rapamycin (Fig 5 f-i). Both experiments, as well as our second mouse model in Fig. 6, support our message that the addition of rapamycin enhances anti-leukemia activity of CAR T cells (STR or LTR) compared to the respective CAR T cells without rapamycin.

In our first revision, to address Reviewer 1, comment 10, we added text that included a comparison of STR with rapamycin and LTR with rapamycin. We have now clarified in the text that this comparison is an observation, not a statistical comparison, and is limited due to product/donor variability, on page 9, paragraph 1:

“Lastly, after showing enhanced anti-leukemic activity in both STR and LTR patient-derived CAR T cells manufactured with rapamycin, in two independent models of leukemia (Nalm6 and 018z), we were intrigued by the possible difference in overall survival of STR with rapamycin compared to the LTR with rapamycin in the Nalm6 model (Fig. 5d, i), with the caveat that a direct comparison of mice treated with STR and LTR is limited due to product/donor variability.”

We respectfully disagree with the reviewer’s statement that the addition of rapamycin to the manufacturing of the LTR product is significantly worsening the antileukemia activity. The data in Fig 5g-i shows the opposite: the addition of rapamycin to the manufacturing of the LTR product significantly enhanced the antileukemia activity, as is stated in the text on page 9, “...we observed

significant reduction in tumor burden (Fig. 5g-h) and prolonged survival in mice injected with rapamycin-treated CAR T cell products from LTR (Fig. 5i).”

We agree that the lower expression of CD69 in STR with rapamycin compared to LTR with rapamycin is complicated, especially given that we observed a difference in survival only in the Nalm6 model. We hypothesize that rapamycin affects the activation status of STR more than LTR, which may explain some of the difference we see in mouse survival in the Nalm6 model. We have clarified this hypothesis in the text on pages 9-10.

“Based on our findings of decreased CAR T cell activation induced by rapamycin (Fig. 4) and the relationship between precise patient-specific control of activation level and CAR T cell functionality²⁰, we hypothesized that rapamycin may have a greater effect on decreasing the activation of STR compared to LTR CAR T cells. STR and LTR CAR T cells without rapamycin had similar levels of CD69 (Fig. 1j). However, we found lower expression of CD69 in STR with rapamycin compared to LTR with rapamycin, including in the patient-derived CAR T cells used in the Nalm6 mice experiments (Supplementary Fig. 8), suggesting that rapamycin decreased the activation of STR CAR T cells to a greater extent than LTR CAR T cells and may explain the difference in mouse survival in the Nalm6 model.”

Lastly, considering the revised version of the paper, we suggest to refine the discussion section:

1) Pointing out that the rapamycin exposure during the manufacturing is transient and limited to day4-6 (line 340).

Response: As proposed by the Reviewer, we clarified in the revised discussion section that rapamycin exposure during the manufacturing process is transient (page 11). We also clarified in the methods section that CAR T cells were treated with rapamycin beginning 4 days after bead stimulation and refreshed throughout CAR T cell manufacturing every other day and not limited to day 4-6 (page 15), as illustrated in Fig. 4a, Fig5a, Fig5f, Fig6a, and Fig6e.

“Based on our studies, we hypothesize that transient treatment with rapamycin, a clinically available drug, offers a simple and rapid metabolic approach to improve current CAR T cell manufacturing practices.”

“Beginning 4 days after bead stimulation, CAR T cells were treated with rapamycin (50 nmol/L) (LC Laboratories, R-5000) or vehicle (DMSO) and refreshed with media containing rapamycin or vehicle every other day throughout manufacturing.”

2) For the CAR T cells exposed to rapamycin, the improved expansion is observed only for the NALM6 model and STR cells (line 335).

Response: We appreciate the recommendation and have revised the manuscript to include more detailed information (page 11).

“Indeed, in a leukemia mouse model, patient-derived mTOR inhibited CAR T cell products showed improved expansion post-infusion (Fig. 5e), which resulted in lower tumor burden and longer survival compared to mice treated with control CAR T cells (Fig. 5-6).”

Reviewer #2

Reviewer #4

The authors have articulated the novelty of the manuscript more clearly and have addressed most of the previous comments. The overall quality of the manuscript has been improved. There are a few minor issues that still require clarification before publication.

1. The authors have clarified the CD27 functions not only as a memory-associated maker but also as an effector-related maker. It's still unclear that if the memory-associated genes play a role in CAR T products from LTR and STR (related to previous comments #1)

Response: As proposed by the reviewer, we highlighted in the revised text (page 5) that memory related proteins do not play a role in CAR T cell products from STR and LTR. Furthermore, Reviewers' 2 second comment regarding labeling the statistical significance proteins further show it.

"We did not observe significant differences in the abundance of different T cell subsets measured using canonical markers CD45RA/CCR7 (Supplementary Fig. 7a-c), as well as the level of CCR7 in CD4 and CD8 CAR T cells (Fig. 1j), suggesting that differences in CAR T cell metabolic functionality between the two sub-groups are not explained by their differentiation or memory markers".

2. Please label the genes that showed statistical significance in Fig. 1j similar to Fig. 4n, to enhance the readability of the figure.

Response: We appreciate the recommendation. As suggested, we modified Figure 1j accordingly.